# Ensemble learning based transmission line fault classification using phasor measurement unit (PMU) data with explainable AI (XAI)

**Simon Bin Akter**[1], **Tanmoy Sarkar Pias**[2], **Shohana Rahman Deeba**[1], **Jahangir Hossain**[3], **Hafiz Abdur Rahman**[1] *

**1** Department of Electrical & Computer Engineering, North South University, Dhaka, Bangladesh,
**2** Department of Computer Science, Virginia Tech, Blacksburg, VA, United States of America, **3** School of Engineering, The University of British Columbia, Vancouver, Canada

* hafiz.rahman@northsouth.edu

**Data Availability Statement:** All data are available without any restriction at the following URL: https://doi.org/10.5061/dryad.x69p8czq3.

## Abstract

A large volume of data is being captured through the Phasor Measurement Unit (PMU), which opens new opportunities and challenges to the study of transmission line faults. To be specific, the Phasor Measurement Unit (PMU) data represents many different states of the power networks. The states of the PMU device help to identify different types of transmission line faults. For a precise understanding of transmission line faults, only the parameters that contain voltage and current magnitude estimations are not sufficient. This requirement has been addressed by generating data with more parameters such as frequencies and phase angles utilizing the Phasor Measurement Unit (PMU) for data acquisition. The data has been generated through the simulation of a transmission line model on ePMU DSA tools and Matlab Simulink. Different machine learning models have been trained with the generated synthetic data to classify transmission line fault cases. The individual models including Decision Tree (DT), Random Forest (RF), and K-Nearest Neighbor (K-NN) have outperformed other models in fault classification which have acquired a cross-validation accuracy of 99.84%, 99.83%, and 99.76% respectively across 10 folds. Soft voting has been used to combine the performance of these best-performing models. Accordingly, the constructed ensemble model has acquired a cross-validation accuracy of 99.88% across 10 folds. The performance of the combined models in the ensemble learning process has been analyzed through explainable AI (XAI) which increases the interpretability of the input parameters in terms of making predictions. Consequently, the developed model has been evaluated with several performance matrices, such as precision, recall, and f1 score, and also tested on the IEEE 14 bus system. To sum up, this article has demonstrated the classification of six scenarios including no fault and fault cases from transmission lines with a significant number of training parameters and also interpreted the effect of each parameter to make predictions of different fault cases with great success.

**Funding:** Energy and Power Research Council (EPRC) of the Government of Bangladesh, Grant # EPRC/58-2018-007-01, PI - Dr. Hafiz Abdur Rahman, North South University. The funders had no role in study design, data collection and analysis, decision to publish, or preparation of the manuscript.

**Competing interests:** The authors have declared that no competing interests exist.

## Introduction

Electrical transmission lines are the key infrastructure that drives modern societies. Identification and isolation of transmission line faults are essential for the smooth operation of electrical infrastructure. The electrical power infrastructure is always susceptible to interruption or an electrical malfunction since it is made up of many complex, dynamic, and interrelated components. Conventionally these faults are identified based on local measurements of voltage and current. However, it is very difficult to maintain the stability and security of large power system networks using only local measurement-based protection. Hence wide-area measurements-based protection mechanisms are developed. It is based on the synchronized sampling of remote buses using Phasor Measurement Units (PMU) and the sample data are processed at the control centers. The processing is done for online security analysis, post-fault analysis, and various other power systems state estimation. Yaser et al. [1] have conducted an in depth analysis of transmission system fault scenarios. This paper discussed several power system fault types and fault analysis methodologies. A power system usually runs in a balanced environment. The power system may have encountered faults for various reasons, including insulation failure, tree fall, bird shorting, and natural occurrences such as lightning, strong winds, and earthquakes. Open circuit faults and short circuit faults are the two main forms of power system disturbances. In this research, only the impacts of short circuit faults of the transmission lines have been covered. When the system becomes unbalanced due to insulation failures at any point or when live wires come into touch with each other, it's claimed that a short circuit fault will occur in the line.

Aspects of power system short circuit scenarios have been thoroughly interpreted by Guangming et al. [2] The fault characteristics of the short circuit instances for the overhead line have been discussed in this paper. An anomalous connection between two sites with different potentials that may happen inadvertently or on purpose has an extremely low impedance, which is known as a short circuit. These are the most common and harmful faults that result in an excessively high current flowing through the equipment of the transmission lines. The equipment will sustain significant damage if these disturbances are permitted to persist for only a short period. Short circuits can be caused by both internal and environmental influences. Internal consequences include failure of any equipment of transmission line, old or faulty installations, poor design, and deteriorating insulation in generators, transformers, and other electrical equipment. External impacts include mechanical damage, equipment overloading caused by lighting surges, and insulation failure. The impacts of short circuit faults can be threatening. Equipment like transformers and circuit breakers can catch fire and explode due to these arcing faults. Equipment that receives abnormal currents becomes overheated, which affects its longevity service. Operational voltages of the power system may fluctuate beyond the acceptable range, which would be detrimental to the quality of electric power delivered. The balanced power flow of a transmission line can be fiercely constrained or stopped due to short circuit circumstances.

Loiy et al. [3] conducted a comprehensive analysis of transmission line fault characteristics. In this paper, the causes and effects of short circuit faults have been discussed. Short circuit circumstances refer to two types of scenarios, such as symmetrical and unsymmetrical. When all three phases are shorted at the same time, a balanced fault case scenario occurs that is referred to as a symmetrical fault that has been illustrated in Fig 1. Three phase faults (LLL) and three phase ground faults (LLL-G) are two instances of symmetrical faults.

The most prevalent kind of faults that a transmission system encounters are asymmetrical circumstances. Currents with varied magnitudes and uneven displacement of phases are instances of asymmetrical faults, which lead to an unbalanced situation. These unbalanced

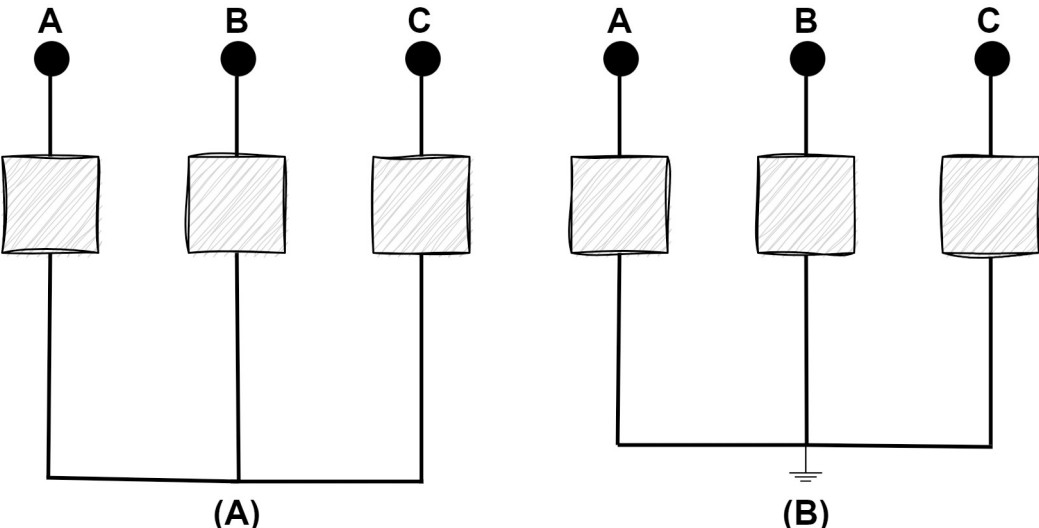

**Fig 1. Observations of transmission lines in terms of symmetrical fault circumstances.** (A) Short circuit between three phase conductors. (B) Short circuit between three phase conductors and ground.

faults are referred to in three scenarios, which have been referred in Fig 2, such as single line to ground fault (L-G), double line fault (LL), and double line to ground fault (LL-G).

A single line to ground (LG) fault, which is the most common fault compared to all electrical system disturbances. When a single line is shorted out with the ground, this scenario refers to single line to ground fault. These are significantly less impactful than other faults. Likewise, a double line (LL) fault occurs when two live wires make contact with one another. Strong winds that might lead to overhead wires swinging into one another are the main cause of this type of fault. Even while these faults aren't as dangerous, these faults may happen a reasonable

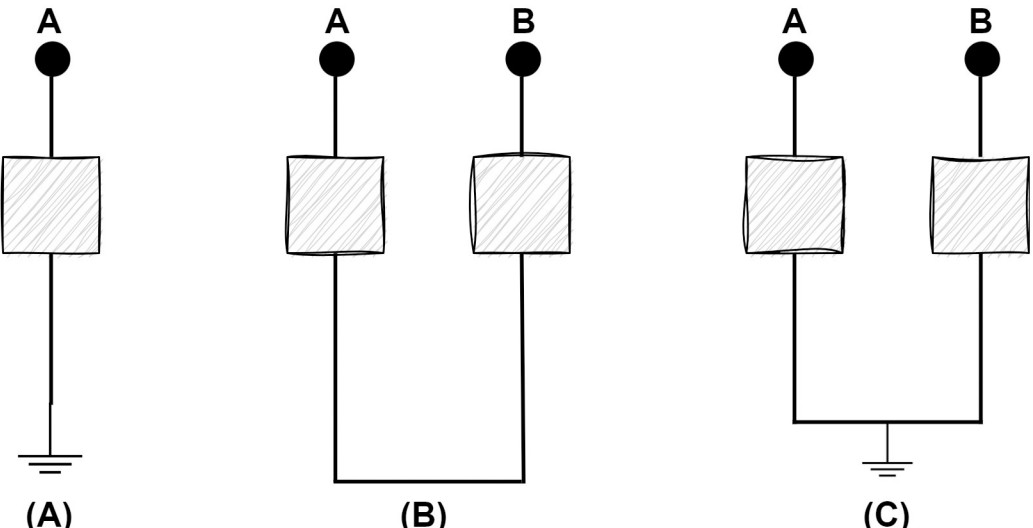

**Fig 2. Observations of transmission lines in terms of unsymmetrical fault circumstances.** (A) Short circuit between a single line and ground. (B) Short circuit between two lines. (C) Short circuit between two lines and ground.

number of times. When two lines are shorted out simultaneously with the ground, this situation leads to double line to ground (LL-G) fault. These severe faults happen seldom as a proportion of total system outages.

Understanding the applications of artificial intelligence (AI) in the power grid can be useful to monitor and maintain the normal power flow in the distribution network. AI models are capable of predicting transmission line faults instantaneously, which ensures effective and early identification of line faults to prevent the consequences.

In light of the existing related works, it is notable that labeled data of different fault cases of transmission lines is limited. To classify different fault cases instantaneously, a preprocessed dataset that contains labeling of each faulty circumstance is crucial. In this research, a simulation based dataset with proper labeling of each short circuit fault occurrence has been generated that overcomes a major challenge to conduct research in this field. Additionally, most research in this field is limited to the estimation of only current and voltage phases. In this research, features have been extracted from the simulation with the Phasor Measurement Unit (PMU) that considers more parameters in accurate fault examination and prediction.

Transmission line fault analysis is very effective for the early identification of different fault case scenarios. To analyze the fault cases, historical data of previous fault scenarios is an indispensable factor. Janarthanam et al. [4] presented an in depth overview of the short circuit fault analysis. The implementation of machine learning techniques in power system fault detection has been analyzed in this paper. To classify short circuit faults using supervised machine learning techniques [5–7], a labeled dataset is required. A transmission line model has been simulated in order to generate a labeled dataset where data for each type of fault scenario has been categorized. Using the generated dataset, classifiers have been trained to build the model for classifying short circuit disturbances of the transmission line. Michael et al. [8] have described the relevance of the use of phasor Measurement Units (PMU) in power systems. This paper provided a practical observation of the implementation of PMUs in power systems to perform power flow analysis. Similarly, in this research Phasor Measurement Unit [9–11] data has been used to retrieve more features to train the models. The developed models have achieved a high accuracy using the generated dataset accordingly. Eoin et al. [12] illustrated the effect of explainable AI in machine learning. This paper discussed black-box analysis [13] of the ML classifiers to identify certain patterns in terms of making decisions. In this research background analysis of the performance of the machine learning models has been evaluated using explainable AI [13–15] to ensure the model's trustworthiness. Similarly, a sensitivity study has been performed to determine the influence of various input domains on the model's predictive capabilities.

The key contributions are summarized below.

1. An ensemble learning based model has been proposed, which achieved cutting-edge accuracy (average accuracy: 99.88%), representing a significant advancement in transmission line fault classification.

2. Explainable AI (XAI) has been applied to analyze the performance of the combined models in the ensemble learning approach that interprets the learning patterns and the decision making processes of the models. This increases the dependability of the results provided by the machine learning algorithms.

3. The study highlights the significance of using Phasor Measurement Unit (PMU) data to increase the accuracy of the developed classification model.

4. The study presents a large volume of multivariate labeled data of transmission lines which opens new opportunities and challenges for addressing supervised approaches to analyze transmission line faults.

5. Principal component analysis (PCA) has been performed on the input features to reduce the noise from the dataset, which ensures quality data for model training.

6. Exploratory data analysis (EDA) has been performed on raw data to visualize and preprocess the dataset which reveals multiple efficacious patterns within the dataset

In this research, our objective is to create an artificial intelligence platform for monitoring the power grid with 4IR [16–19] technologies. The developed model is considered capable of identifying and categorizing short circuit events in real-world circumstances. It has been planned to assign the developed model to the different points of the physical transmission line so that details of any blackouts or other unusual events that may occur in the electrical grid can be accessed.

## Literature review

System instability, equipment damage, and even power outages can all be brought on by power system faults. Therefore, power system reliability and consistency depend on the early and precise detection and categorization [20–23] of power system faults. The PMU data has widespread usage in power system fault classification [24–26] engaging machine learning techniques. The objective of this literature review is to give an overview of recent work on fault classification employing machine learning methods.

In 2021, Praveen et al. [27] presented a customized Convolutional Neural Network (CNN) for fault classification in distributed networks combined with distributed generators (DGs). The proposed approach beat traditional methods in terms of accuracy and computational burden, with an average 10-fold cross-validation accuracy of 99.52% for all evaluated fault instances. Accordingly, a mixed transmission line and distribution network using photovoltaic (PV) as the DG was used to evaluate the proposed method, and the results showed performance accuracy of 99.92% and 99.97%, respectively.

Anurag et al. [28] developed a customized Convolutional Neural Network (CNN) with automatic feature extraction to classify power system faults. The suggested method avoids any pre-processing by immediately feeding CNN with the voltage and current signals' raw data. The model was evaluated on an IEEE-30 bus system using a 10-fold cross-validation approach, obtaining an average accuracy of 99.27%. Multi-class fault classifications had been performed using the proposed model including the no fault scenario, which exhibited great efficiency in terms of a number of performance metrics.

Mostafa et al. [29] presented an article based on Latent Structure Learning inside a multi-task learning framework that improves learning and decreases overfitting to unlabeled data. The proposed model has been applied to classify power distribution system faults. An experimental analysis was used to validate the suggested approach and show its robustness in terms of identifying the noise. The validation score was 99.02% as a response.

Zikang et al. [30] proposed a Univariate Temporal Convolutional Denoising Autoencoder (UTCN-DAE) to encode and decode univariate disturbance data using a Temporal Convolutional Network. The paper proposed a rapid and trustworthy method for classifying data quality-related disturbances in practical phasor measurement devices (PMUs). The method achieved optimal feature extraction of multivariate time series by merging features using a two-stream augmented network that had successfully tolerant poor data. The data was classified using a softmax classifier and a deep neural network with many layers. As evidenced by exhaustive results on the IEEE 39-bus system and a sizeable power system in China using field PMU observations, the suggested method provided considerable classification accuracy and computational economy. The total test accuracy was 97.69%, showing

that the approach performed outstandingly to classify disturbances in terms of poor-quality data.

Mohammed et al. [31] proposed an anomaly based method for fault identification employing One-Class Support Vector Machine (SVM) and Principal Component Analysis (PCA) models. This study analyzed the difficulty of fault detection in electrical power systems. The VSB power line failure detection dataset was used to train and test the models. According to the experimental findings, the proposed method performed significantly in fault identification, with accuracy for the two models approaching 80%.

In 2021, Ibrahim et al. [32] introduced a novel method for fault diagnosis based on dissolved gas analysis (DGA) by integrating six optimal machine learning (OML) approaches with data processing. The six OML techniques that had been employed include Decision Trees, Discriminant Analysis, Naive Bayes, Support Vector Machines, K-Nearest Neighbors, and Ensemble classification techniques. In the study, 361 samples were utilized for training purposes and 181 samples were used for testing, which included a total of 542 dataset samples obtained from libraries and labs. The research emphasized how machine learning could enhance DGA's power transformer fault diagnostic capabilities. The Ensemble approach achieved a prediction accuracy of 90.61% for testing dataset samples.

The network proposed by Pullabhatla et al. [33] required fewer layers and fewer data to achieve high accuracy while eliminating the issue of overfitting. The paper introduced a unique method for categorizing power system issues using a 3D deep learning algorithm. The input to the network was a 4D picture created by converting fault currents in an IEEE-9 Bus system to a time-frequency domain and then dividing the resulting matrix into RGB color channels. The proposed model had been trained many times with different epochs and learning rates, and with a training-validation sample size of 1600 4D-images for the suggested dropout values, it achieved an accuracy of 93.75% and 100% for the dropout values of 0.4 and 0.5, respectively.

Shahriar et al. [34] proposed a model that was successful in operating with a small amount of data to provide good accuracy. This approach efficiently extracted key fault features from a small amount of data, which played a significant role in improving model performance. In this article, a capsule network with sparse filtering (CNSF) is presented as the basis for an unsupervised framework for transmission line fault detection and classification. In the suggested method, post fault three phase signals were combined into a single image, which was then used as the input for the suggested CNSF model. In order to prove the model's high dependability, its performance against noise, high impedance faults, and line parameter fluctuations was also been evaluated. The suggested CNSF model achieved an accuracy of 99% against noise and more than 97% against high impedance faults and variations in line characteristics.

In 2021, Fezan et al. [35] introduced a novel machine learning approach for recognizing and classifying power transmission network faults. The suggested method eliminated the necessity for feature extraction by using Long Short-Term Memory (LSTM) units to learn directly from the temporal sequence of operational data. It had been found that the approach was quick and robust after testing it for numerous fault kinds, under various circumstances, and with diverse noise levels. A WSCC 9 bus system was used to test the suggested technique. Based on 5000 samples of test data, the model's combined classification accuracy was 99%.

To detect events in power networks, Zikang et al. [36] proposed a synchrophasor measurement based strategy that accounted for varying amounts of renewable energy penetration. The approach was comprised of an integrated Additive Angular Margin (AAM) loss and Long Short-Term Memory (LSTM) network for feature extraction in order to manage intra-class similarity and inter-class variation. For adaptive data window selection and rapid event pre-classification, Multi-stage Weighted Summing (MSWS) loss-based criteria were devised.

Hence, a model for re-identification based on feature similarity was also developed. Simulation findings on the IEEE 39-bus, two-area Kundur's model, and a genuine large-scale power grid system indicated the benefits of the proposed technique in comparison to competing ones. The test accuracy of 97.99% specified that the suggested technique had performed outstandingly for event detection in terms of renewable energy integration.

R. Machlev et al. [37] suggested an open-source application that could produce power quality disturbance (PQD) synthetic data, which was created particularly for comparing PQD classifiers. The article highlights the necessity of PQD detection and classification as well as the increasing demand for power quality monitoring technologies. The system was built with two reference classifiers based on deep learning. It also incorporated a variety of standard disturbances with variable repetitions and random parameters. These classifiers, according to the authors, could serve as benchmarks for the creation of new and enhanced PQD classification methods. For CNN with and without noise, the test accuracy was 99.75% and 99.28%, respectively. Similarly, for BiLSTM with and without noise, the test accuracy was 98.13% and 96.289%, respectively.

Zain et al. [38] suggested three multi-level deep learning techniques for detecting, classifying, and localizing SS, SG, and OC faults based on Convolutional Neural Networks (CNN), Long Short-term Memory (LSTM), and Bi-directional Long Short-term Memory (Bi-LSTM) networks. The proposed strategy reduces by 50% the number of sensors needed per string. A noisy dataset was introduced to the suggested approach to evaluate the effectiveness of the model. The proposed model obtained 99.94% and 99.54% accuracy correspondingly for fault classification and localization.

In order to automatically identify and pinpoint symmetrical and asymmetrical faults, as well as high-impedance faults (HIFs), in a distribution system, Jibin et al. [39] suggested an end-to-end deep learning technique that made use of a unique Deep Convolutional Neural Network Transformer model. The suggested model made use of 1-D deep CNNs for feature extraction, transformer encoders for sequence learning, and an attention mechanism to isolate key time intervals and draw out the context of the temporal current data. The suggested Xception Transformer (XT) model had outperformed traditional fault detection methods, according to the outcome of simulating various faults using MATLAB Simulink. Prediction accuracy for fault classification and localization were respectively 97.53% and 96.14%.

According to recent studies illustrated in Table 1, ML techniques [40–43] are quite accurate in categorizing various fault types. Real-time labeled datasets [37] of power system disruptions are rarely present, hence synthetic data has played a vital role in these fields. Similarly, power system fault case simulations [30, 39] have been a valuable asset for testing classification models in different fault circumstances. It is necessary to thoroughly examine the backgrounds of AI models before implementing them for real-time power system monitoring, which is currently lacking in the field of study. The training procedures of these AI models resemble black boxes since ensuring transparency regarding the learning patterns and decision making processes becomes an imperative factor to employ them as a monitoring tool in the actual power grid. Similarly, most studies of transmission line fault classification using AI models are limited to the analysis of only current and voltage estimations. This is the reason for the lack of utilization of the Phasor Measurement Unit (PMU) based data acquisition techniques in the field of study. Phasor Measurement Units (PMU) devices are used for wide-area monitoring and protection of electrical transmission lines. The PMUs use synchronized sampling of power grid buses (voltage and current) which are processed at control centers for online security analysis, post-fault analysis, and various other power systems state estimation. PMU based approach generates a large amount of data, which is appropriate to be analyzed using AI techniques of transmission line fault classifications. An open access labeled dataset with a

**Table 1. Overview of the literature review.**

| Researcher | Year | Research Topic | Dataset | Modeling Technique | Performance |
|---|---|---|---|---|---|
| Jibin et al. [39] | 2023 | Power system networks fault classification and localization | Multi-label | XT | 97.53% (classification) 96.14% (localization) |
| Zikang et al. [30] | 2022 | Power system disturbance classification | Multi-label | TCN | 97.69% |
| Zain et al. [38] | 2022 | Photovoltaic systems fault classification and localization | Multi-label | CNN | 99.94% (classification) 99.54% (localization) |
| Praveen et al. [27] | 2021 | Distribution network fault classification | Multi-label | CNN | 99.52% |
| Anurag et al. [28] | 2021 | Power system fault classification | Multi-label | CNN | 99.27% |
| Mostafa et al. [29] | 2021 | Power distribution system fault classification | Unlabeled | MTLS-LR | 99.02% |
| Mohammed et al. [31] | 2021 | Power system networks fault detection | Binary-label | SVM PCA | 79.84% 79.28% |
| Ibrahim et al. [32] | 2021 | Power transformer fault diagnosis | Multi-label | EL | 90.61% |
| Pullabhatla et al. [33] | 2021 | Power system fault classification | Multi-label | DLA | 93.75%—100% (dropout: 0.4—0.5) |
| Shahriar et al. [34] | 2021 | Transmission line fault detection and classification | Unlabeled | CNSF | 97%—99% (noise—high impedance) |
| Fezan et al. [35] | 2021 | Transmission line fault detection and classification | Multi-label | LSTM | 99.00% |
| Zikang et al. [36] | 2021 | Power system event identification | Multi-label | LSTM | 97.99% |
| R. Machlev et al. [37] | 2021 | Power quality disturbances dataset generator with reference classifiers | Multi-label | CNN BiLSTM | 99.28%—99.75% (without noise—with noise) 96.28%—98.13% (without noise—with noise) |

significant number of parameters of transmission line fault circumstances will overcome the major limitations in the field of study.

## Dataset

This time series dataset includes data on various electrical grid scenarios, including no fault and short circuit occurrences. Several fault case scenarios have been simulated using ePMU DSA tools [44] and Matlab Simulink [30, 39] to produce this dataset. To accumulate data, the Phasor Measurement Unit (PMU) has been positioned on the transmission line simulation model. It is not practical to create actual faults in an existing power grid to accurately account for erroneous realtime data as simulating faulty scenarios of power systems has been an effective approach to work in this field.

Based on the level of contaminants, features are ranked as the most important features. Green indicates raw features from the Phasor Measurement Unit (PMU) and blue indicates current and voltage phases. The Phasor Measurement Unit (PMU) provides information on the frequency, phase angle, and magnitude of the current and voltage phases. It is notable that the prediction models have been successfully trained using the data of the Phasor Measurement Unit (PMU). From the Fig 3, it can be stated that the predicting performance of the models has benefited significantly by incorporating PMU data rather than just using the value of current and voltage phases.

The complete overview of the entire dataset has been presented in Table 2. Initially, the dataset contains 134402 rows and 14 columns without preprocessing. There are 12 features in the dataset, including frequencies and phase angles, magnitudes as well as values of the currents and voltages of three phases.

In Table 3, the complete overview of the features retrieved from the Phasor Measurement Unit (PMU) has been described. The dataset contains an estimation of current and voltage

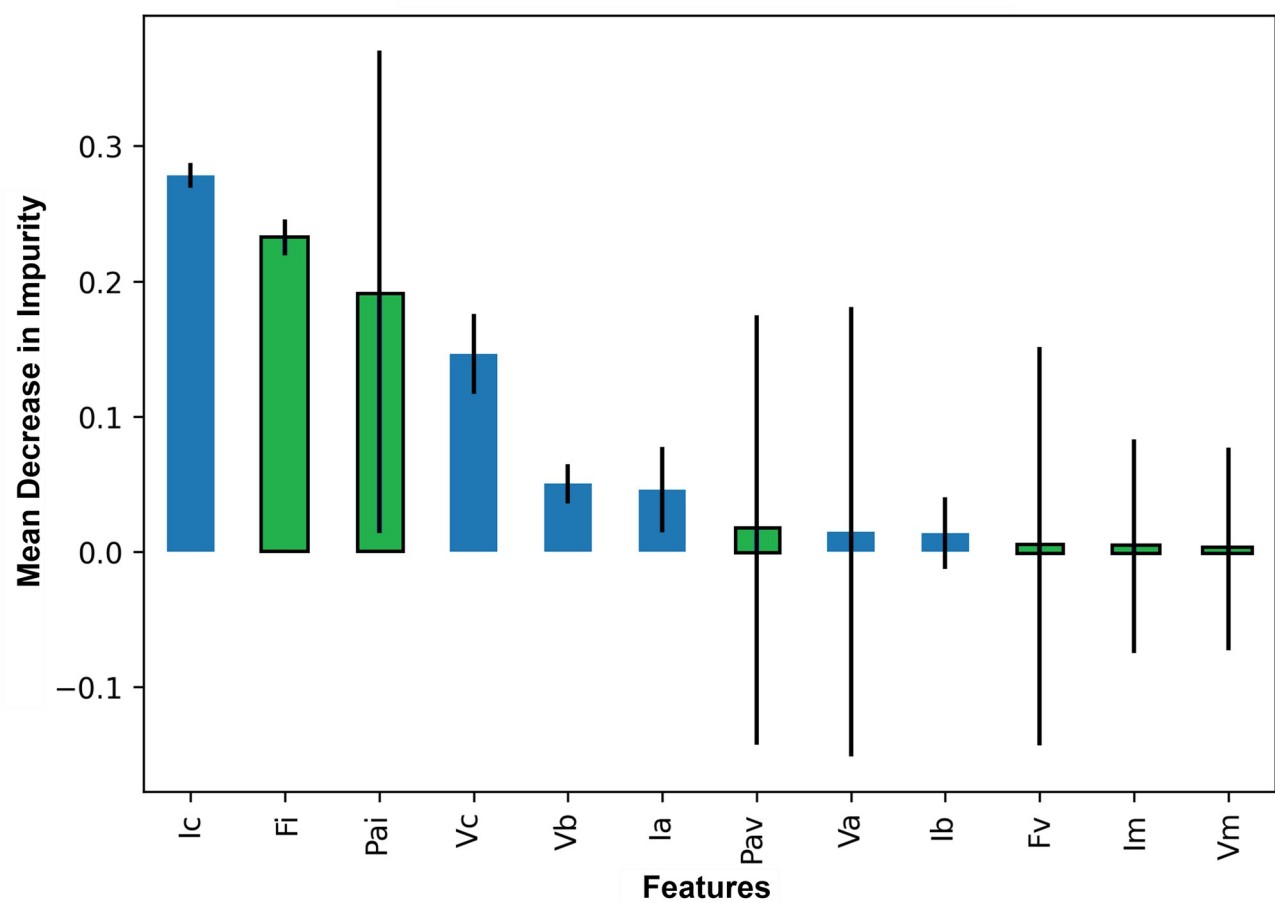

**Fig 3. Feature importance of the extracted characteristics from the simulation using Mean Decrease in Impurity (MDI).**

**Table 2. Overview of the dataset.**

| Types of dataset | Multivariate |
|---|---|
| Number of rows | 134402 |
| Number of columns | 14 |
| Features | timestamp, Ia, Ib, Ic, Va, Vb, Vc, Im, Vm, Fi, Fv, Pai, Pav |
| Labels | NF, L-G, LL, LL-G, LLL, LLL-G |

**Table 3. Overview of feature columns.**

| Feature | Description |
|---|---|
| Ia, Ib, Ic | Three-phase sinusoidal current. |
| Va, Vb, Vc | Three-phase sinusoidal voltage. |
| Im, Vm | Magnitude of the three phase sinusoidal current and voltage. |
| Fi, Fv | Frequency (Hz) of three phase sinusoidal current and voltage. |
| Pai, Pav | Phase angle (degree) of three phase sinusoidal current and voltage. |

**Table 4. Overview of target labels.**

| Target Label | Description |
|---|---|
| NF | Balanced signal flow, no direct contact between conductors and ground. |
| L-G | A single line fault occurs when one conductor on a transmission line contacts the neutral conductor or drops to the ground. |
| LL | Double line fault refers to a situation in which two conductors are short circuited. |
| LL-G | A double line to ground fault occurs from a short circuit between two phase conductors and ground. |
| LLL | A three phase short circuit fault defines a short circuit between three phase conductors. |
| LLL-G | A short circuit between three phase conductors and the ground is referred to as a three phase to ground fault. |

phases, the magnitude of current and voltage phases, frequency, and phase angle across both current and voltage phases. For classifying no fault and fault case circumstances, the dataset includes six labels, which have been illustrated in Table 4, including no fault (NF), line to ground (L-G) fault, line to line (LL) fault, line to line to ground (LL-G) fault, line to line to line (LLL) fault, and line to line to line to ground (LLL-G) fault.

The dataset has been divided into six portions according to each simulated circumstance. Accordingly, phases for each simulated circumstance have been visualized. In terms of no fault (NF) circumstances, ideal current and voltage flow cycles have been notable. When a fault occurs in the system, currents, and voltages start to depart from the ideal state. Hence, disturbances in the voltage and current phases have been evident in terms of fault circumstances in the system. Current phases have increased extensively and voltage phases have decreased from the initial state which has been presented in Figs 4 and 5.

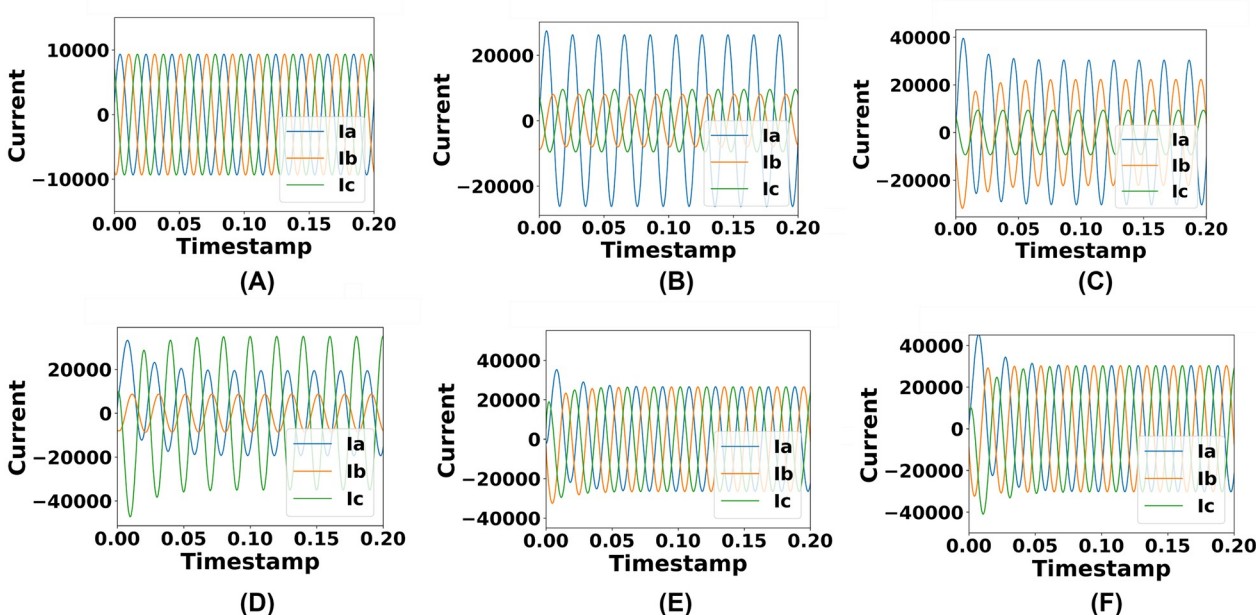

**Fig 4. Representation of the identified changes in current phases in terms of different simulated circumstances.** (A) Current phases under no fault condition. (B) Current phases under single line fault condition. (C) Current phases under double line fault condition. (D) Current phases under double line to ground fault condition. (E) Current phases under three phase fault condition. (F) Current phases under three phase to ground fault condition.

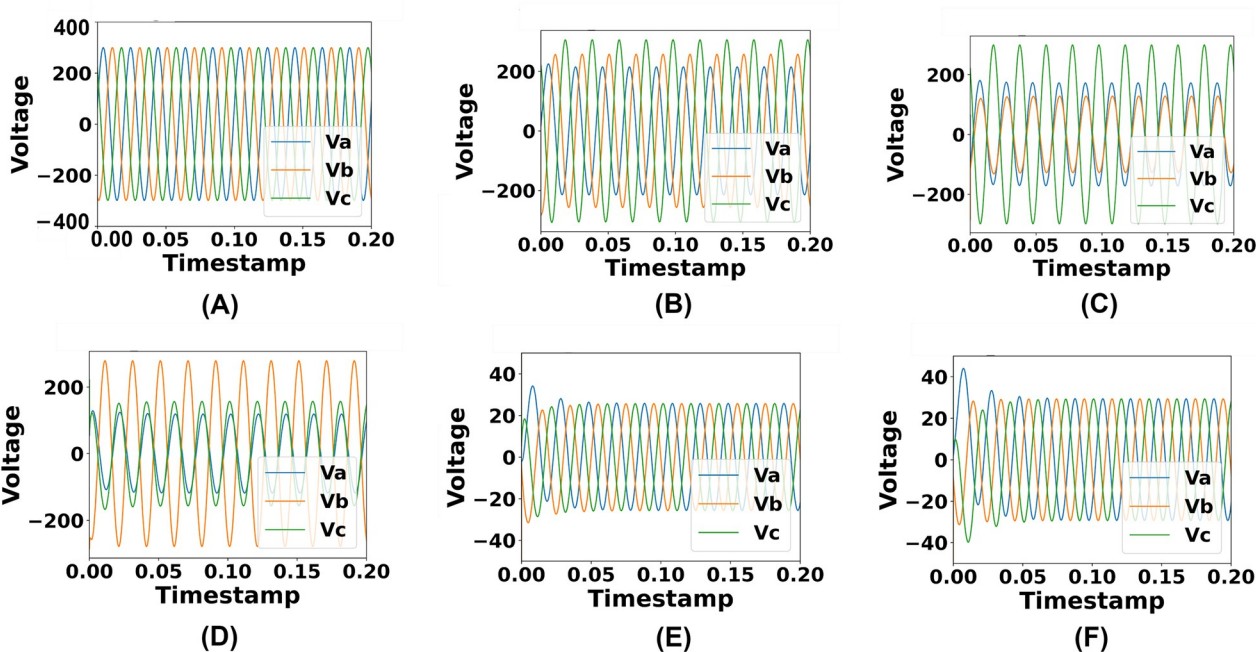

**Fig 5. Representation of the identified changes in voltage phases in terms of different simulated circumstances.** (A) Voltage phases under no fault condition. (B) Voltage phases under single line fault condition. (C) Voltage phases under double line fault condition. (D) Voltage phases under double line to ground fault condition. (E) Voltage phases under three phase fault condition. (F) Voltage phases under three phase to ground fault condition.

When a short circuit happens, voltage has fallen while power has remained constant and the current has increased, which has been visualized in this section. Short circuits with extremely low impedance have caused the high rise of current flow. There hasn't been any resistance or reactance in terms of short circuit scenarios. This illustrates the data variations between the no fault and fault case occurrences.

## Methodology

Synthetic data generation of the transmission line has been conducted using ePMU DSA tools and Matlab Simulink. Generated data from the simulation has been converted into CSV format with 12 feature columns with timestamps and 1 target label. Using Google Colab, Python, and its libraries including Scikit-learn, NumPy, Pandas, SHAP, Matplotlib, and Seaborn exploratory data analysis (EDA) has been performed to visualize and clean the dataset. After preprocessing the data, nine features have been selected from the dataset to train the models. Then the dataset has split into train and test sets. The complete workflow to develop an ideal model to classify transmission line faults has been depicted in Fig 6.

Using the prepared training dataset, data has been trained using supervised multiclass classifiers [40]. The performance of the model to classify six predefined labels has been evaluated by several performance metrics. Accordingly, Cross-validation [28] has been performed to check if the model leads to overfitting or not. 10-fold cross-validation has been selected since it has improved stability while assessing the performance of the model compared to less fold. Similarly, the sensitivity of the predicted performance to the particular data has decreased for the increased number of folds. Hyperparameter tuning has been performed on the best performing classifiers using random and grid search approaches to identify optimal

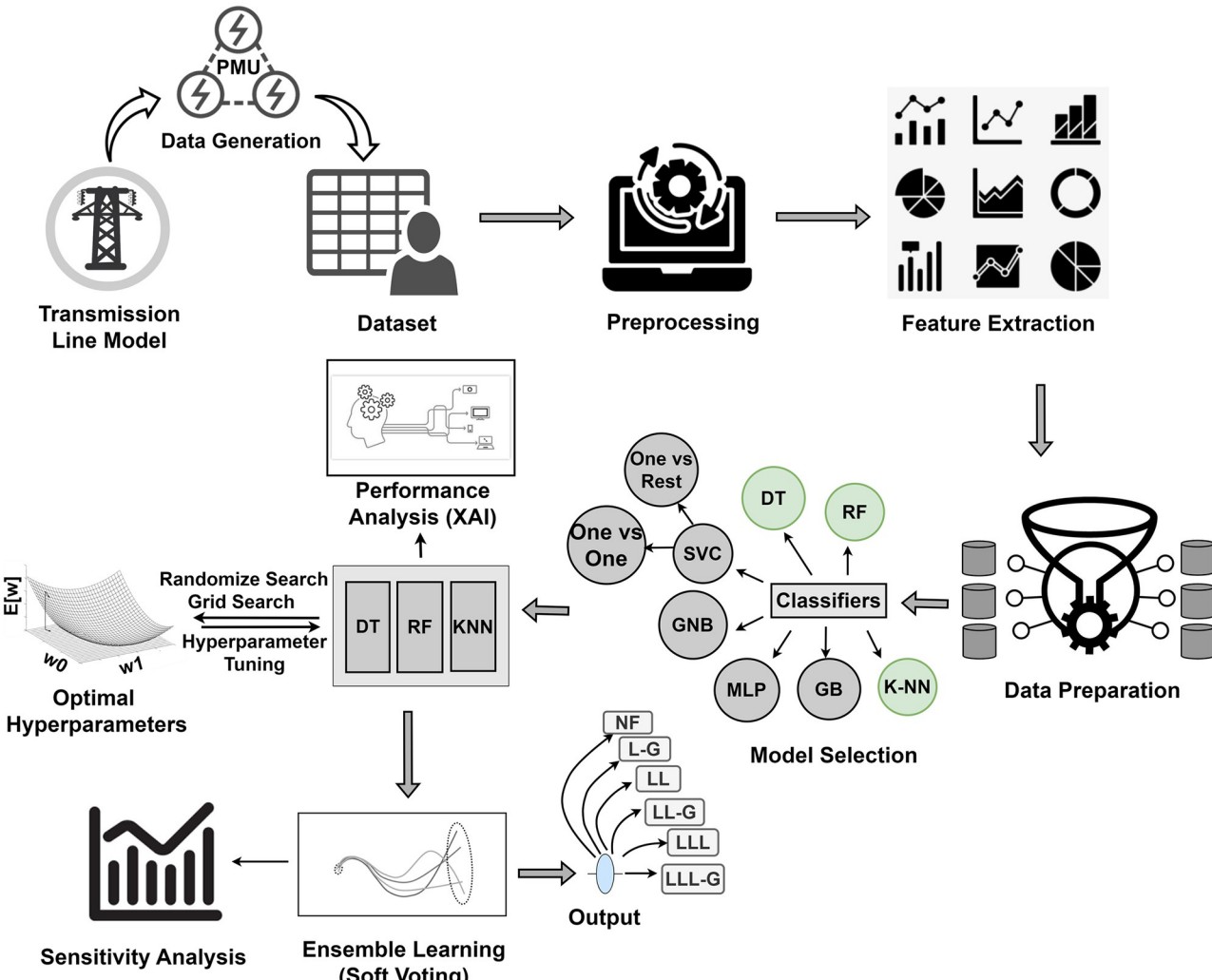

**Fig 6. The Complete workflow of the proposed transmission line short circuit fault classification and exploration through generating synthetic labeled data.**

hyperparameters for these classifiers. Similarly, performance analysis of the best performing models has been overseen using explainable AI [45–47] to ensure the solidity of the classification models. Sensitivity analysis has been performed to determine the manner in which various input domains affected the model's ability to predict outcomes. Consequently, soft voting [48] has been performed to combine the performance of the effective classifiers, in this way an ensemble learning based model has been developed to classify transmission line short circuit disturbances.

## Data generation

A transmission line model was simulated using ePMU DSA tools and Matlab Simulink to produce synthetic data, which have been structured in Fig 7. In the simulation model, a generator having reactive power of 10 MVA connected to a delta-star transformer of 10 MVA reactive power as well as a load that is consuming 5 MW power is also connected with the transmission

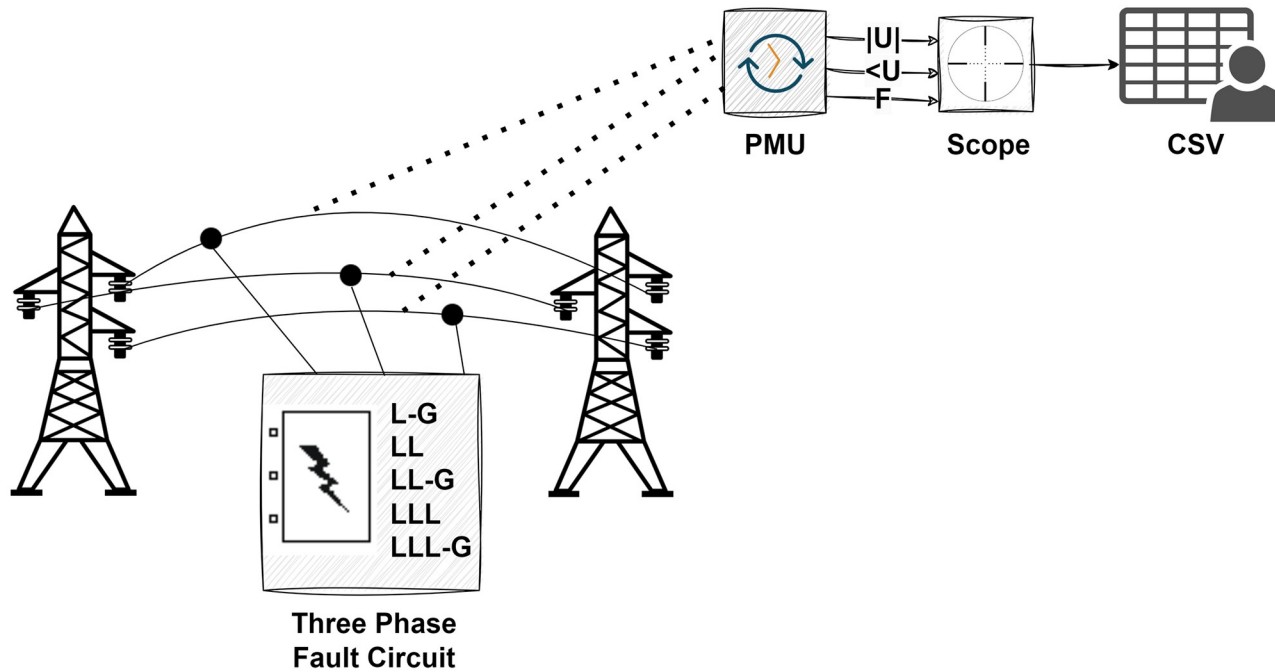

**Fig 7. Representation of synthetic data generation phases for the transmission line.**

line. Accordingly, a fault circuit that has the ability to create short circuit fault scenarios has been attached to the transmission line model. A Phasor Measurement Unit (PMU) has been integrated into the transmission line to collect data. The model has run for 7 seconds for no fault and each fault case circumstance. For each case scenario, data has been collected with timestamps.

Six power system circumstances including no fault and fault cases have been simulated. The Phasor Measurement Unit (PMU) has accumulated data on frequencies, phase angles, and magnitudes of current and voltage phases for each simulated circumstance. The collected data has been converted into CSV format to prepare for preprocessing.

## Preprocessing

In order to clean the dataset, duplicates and null values are initially handled. The value of current and voltage phases varies from no fault to fault scenarios. From the Fig 8, it is noticeable that in terms of no fault cases, the value of current and voltage phases are normally distributed.

The skewness and kurtosis for no fault circumstances represent normal distribution for current and voltage phases, which is in the -2 to 2 and -7 to 7 range respectively. In no fault circumstances, data closer to the mean occur more frequently than data further from the mean. In light of this, it can be said data are symmetric around the mean under no fault conditions. Accordingly, values of current and voltage phases are not normally distributed in terms of fault cases. As current has increased extremely and voltage has decreased in fault cases, skewed values are noticeable. The skewness and kurtosis for fault circumstances represent that data is not normally distributed for current and voltage phases. Faulty data are not symmetric around the mean, indicating that data have a flatter or steeper dome.

The dataset has been divided into two partitions including no fault and faulty then outliers have been identified from the faulty section, which has been presented in Fig 9. As a

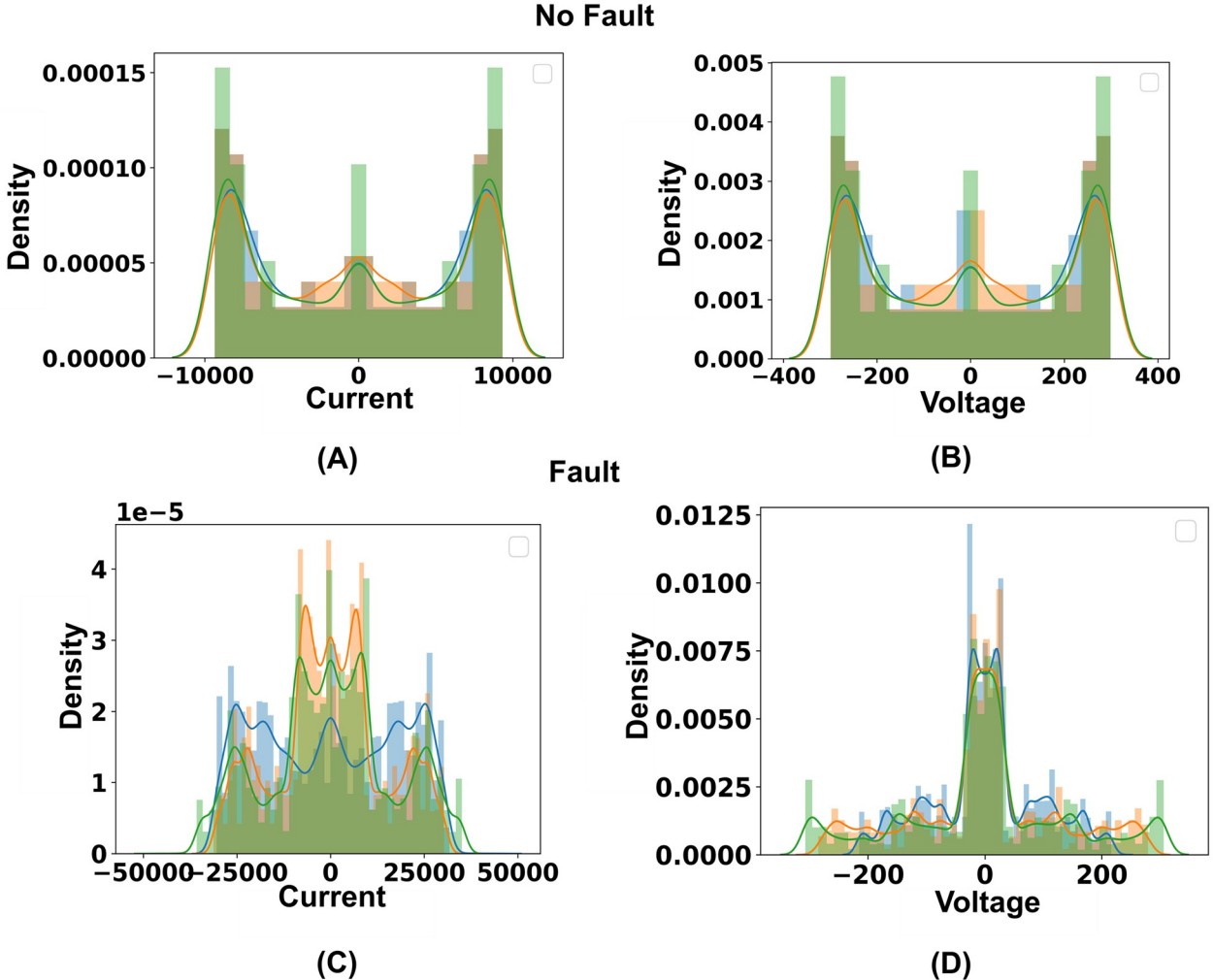

**Fig 8. Representation of the data distribution in current and voltage phases across no fault and fault circumstances.** (A) Data distribution of current phases under no fault condition. (B) Data distribution of voltage phases under no fault condition. (C) Data distribution of current phases under different fault conditions. (D)Data distribution of voltage phases under different fault conditions.

consequence, outliers have been handled using the interquartile range (IQR) approach from the faulty data to reduce noise from the dataset. In this process, observations that have been located more than 1.5 times below from first quartile or 1.5 times above from third quartile have been considered outliers and removed from the dataset.

## Feature extraction

On the transmission line simulation model, a phasor measuring unit (PMU) has been placed. PMU [1, 9, 10] monitors the current and voltage phasor values. Once the simulation model has run, features have been retrieved from the located PMU. In this power system analysis, PMU has extracted values for key characteristics such as magnitudes of current and voltage phases along with phase angles and frequencies. The PMU measured the frequency, phase, and amplitude of the positive sequence component of the given simulation model of electrical signals in a three phase system using a specified sample rate.

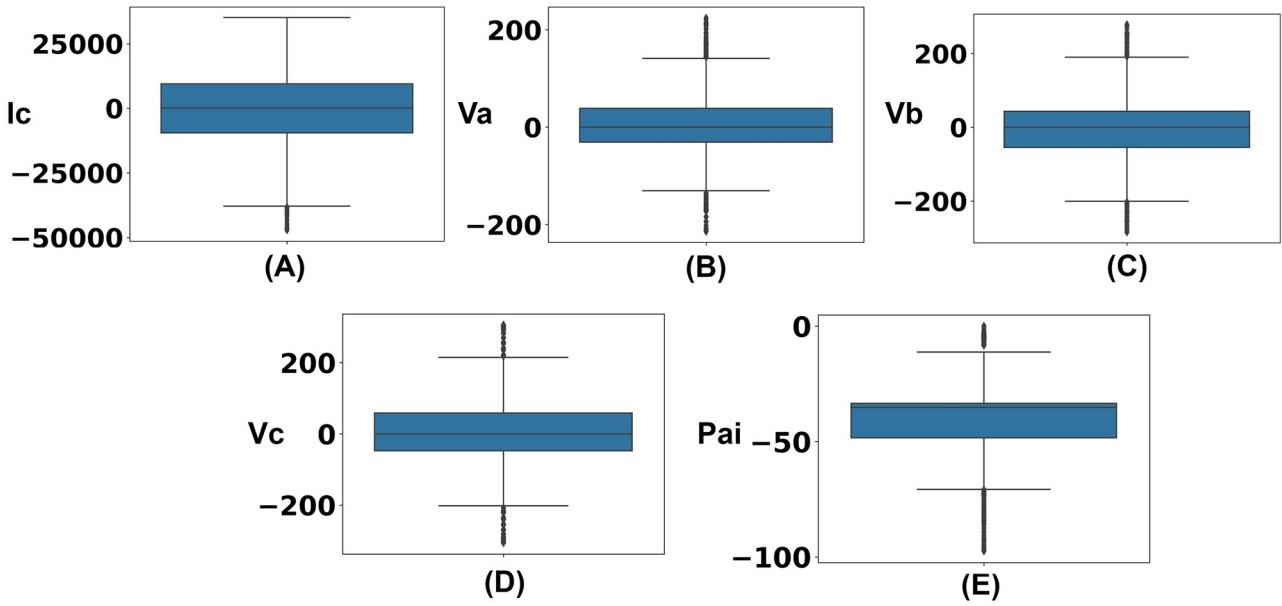

**Fig 9. Visualization of the outliers in terms of fault circumstances.** (A) Outliers of current phase (Ic) in terms of faulty scenarios. (B) Outliers of voltage phase (Va) in terms of faulty scenarios. (C) Outliers of voltage phase (Vb) in terms of faulty scenarios. (D) Outliers of voltage phase (Vc) in terms of faulty scenarios. (E) Outliers of phase angle (Pai) in terms of faulty scenarios.

The complete structure of the feature extraction has been visualized in Fig 10. The dataset contains 12 numeric columns with timestamps and 1 categorical column after data has been retrieved from the PMU. Strongly correlated features are linearly dependent and have the same impact on the dependent variables, which might cause the model to give biased and wrong decisions. It is noticeable that the magnitudes of the voltages are strongly correlated with phase angles. Therefore, the feature magnitude of voltage and phase angle will have the same impact on the dependent variables. Hence, the feature magnitude of three phase voltages has been removed from the dataset. As the value of the three phase currents is present in the dataset, the magnitude of three phase currents is also disregarded. Similarly, phase angles of current and voltage phases also have high correlation rates so the feature phase angle with respect to voltage phases has also been removed from the dataset.

From Fig 11, it is notable that features Ia, Ib, Ic, Va, Vb, Vc, Fi, Fv, and Pai have less correlation among themselves than the other features of the dataset. As such, these features have been chosen as input features to train the model.

## Data preparation

After preprocessing and feature extraction, the dataset was partitioned into two parts, including input and target, to allocate the amount of data from each simulated scenario that would be utilized to train and test the model. The complete approach of the data preparation has been represented in Fig 12. Following that, the dataset was divided into train and test portions. The training dataset contains 107000 rows of data. Accordingly, there are 26881 rows of data in the test dataset. Data from each simulated case scenario has been presented in both train and test sets. In order to make predictions on the target column, label encoding has been used to transform the categorical target column into a numeric column.

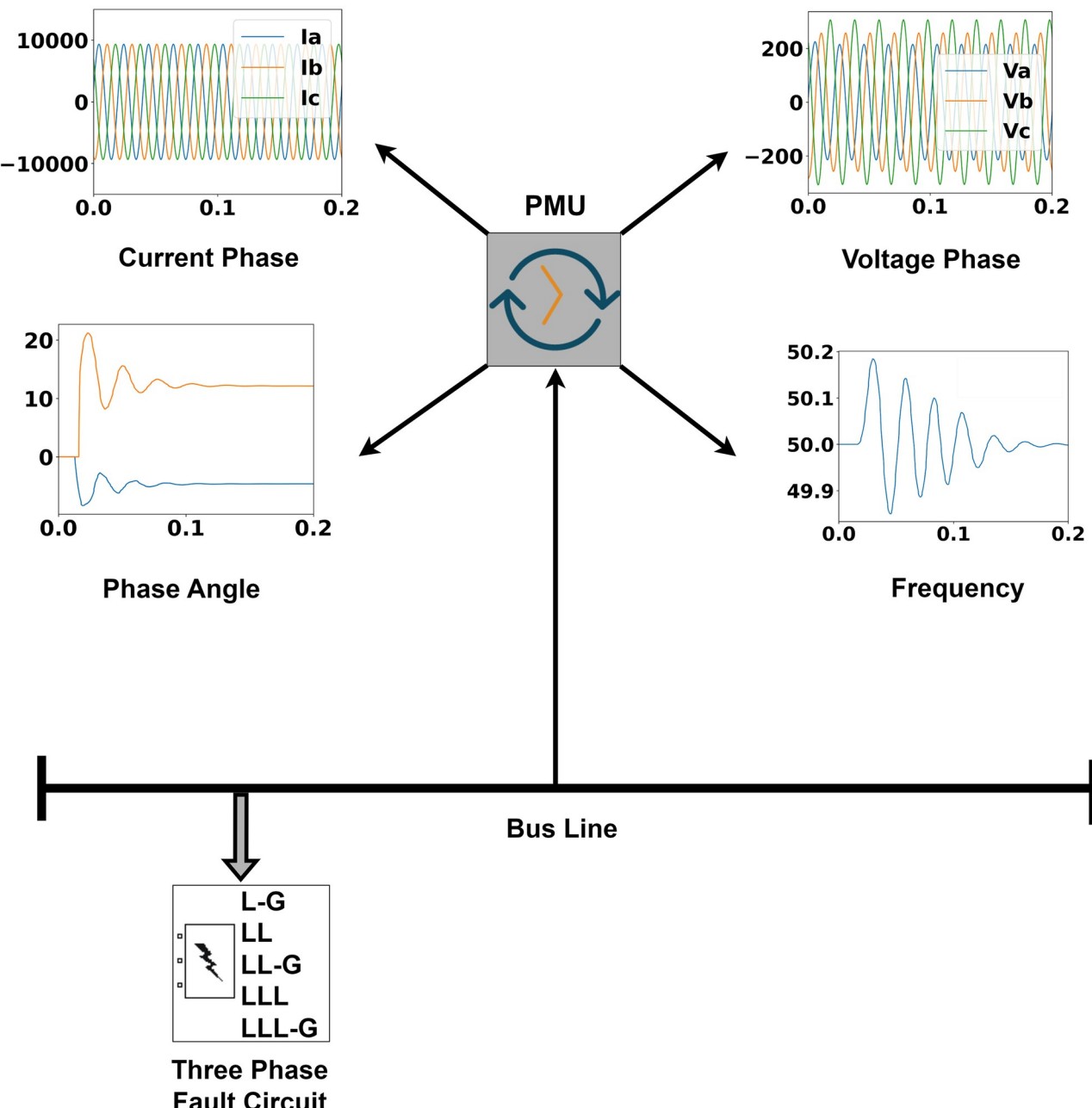

**Fig 10. Representation of feature extraction phases from the generated synthetic data through Phasor Measurement Unit (PMU).**

Accordingly, Principal Component Analysis (PCA) [49] has been performed on the input features of train and test sets to reduce the noise and correlation from the data to improve the performance of the model. It has reduced feature space dimensionality to improve the accuracy of classification models. It has been operated by creating a new collection of features known as principal components (PCs) from the given features of the dataset. In this way, the data has become ready to be employed by multiclass classifiers to train the classification models.

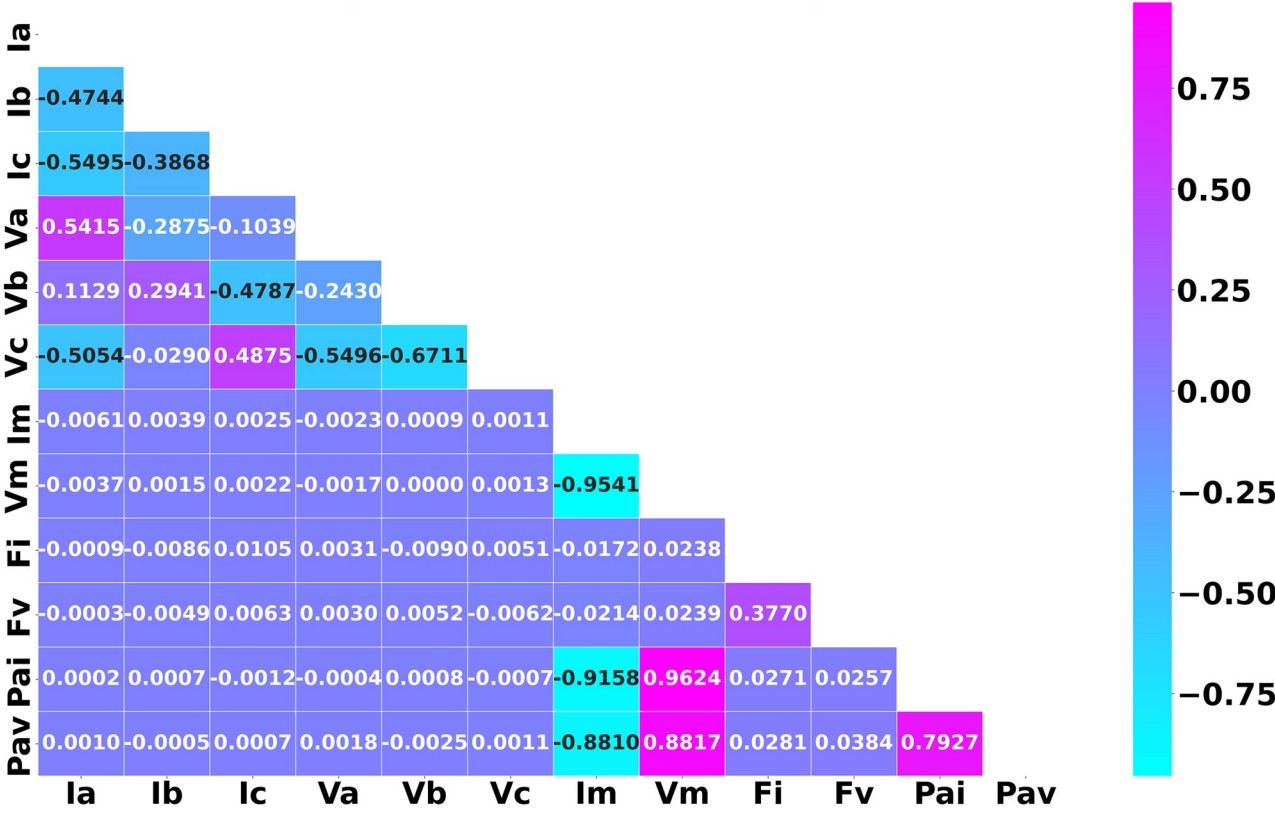

**Fig 11. Representation of the correlation matrix for the extracted features from the transmission line simulation.**

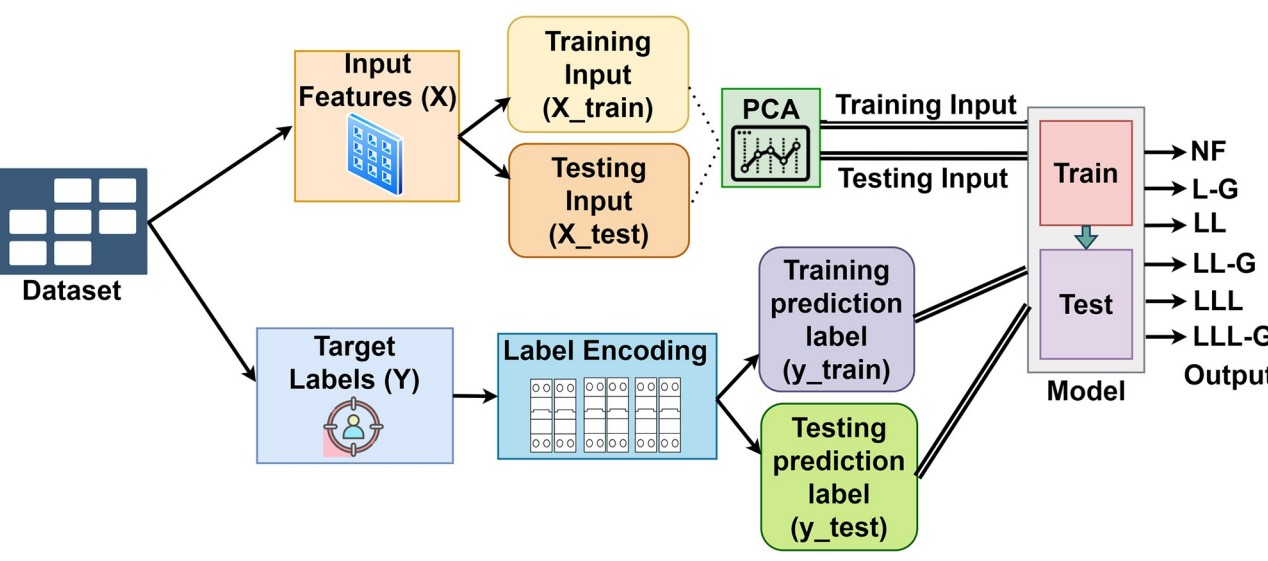

**Fig 12. Representation of the data preparation phases for the selected input features.**

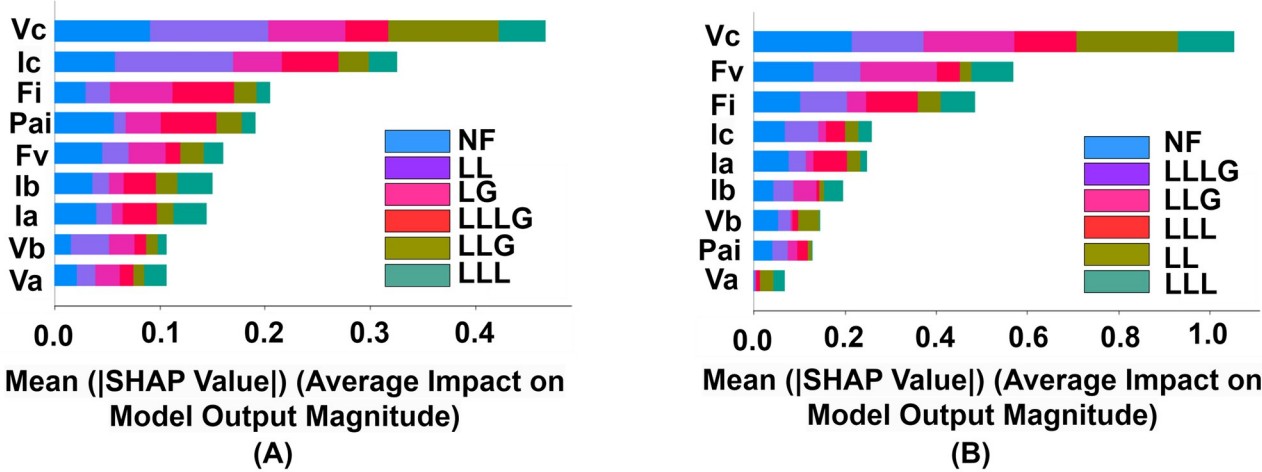

**Fig 13. Visualization of the identified impact from each feature indicating target labels using SHapley Additive exPlanations (SHAP) values.** (A) Influence of each feature for the Decision Tree (DT) model. (B) Influence of each feature for the Random Forest (RF) model.

## Performance analysis

Explainable AI approaches using SHapley Additive exPlanations (SHAP) have been implemented on the best performing models, which provides a deep analysis of the performance of these machine learning models. SHAP [50–52] analysis of a model shows how significant each observation is in deciding the model's final prediction. This approach explains the background functionalities of the models to make predictions by conducting several predictions and then analyzing the outcomes of the prediction against the impact of other factors.

This analysis in the Fig 13 represents how each feature significantly impacts the classification of each class. To classify a few classes, particular features have hardly been used. On the other hand, the same features have been used uniformly for the prediction of another few classes. This represents a high variation rate between these classes.

The inspection from Fig 14 has included feature significance and relevance. Each dot on the summary plot represents an instance of a feature by using Shapley values. This representation

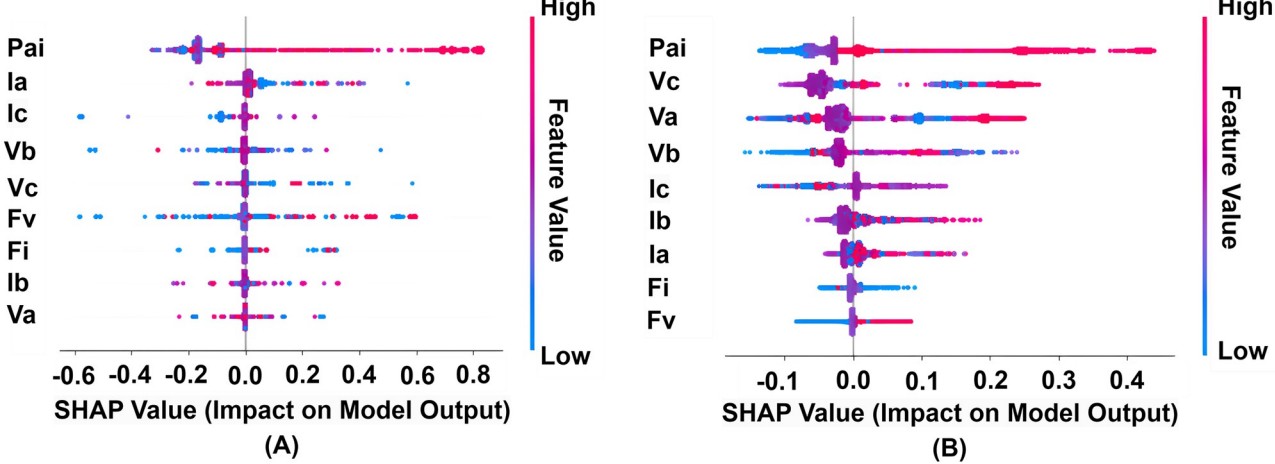

**Fig 14. Visualization of the identified observations from each feature using SHapley Additive exPlanations (SHAP) values.** (A) Influence of each observation for the Decision Tree (DT) model (B) Influence of each observation for the Random Forest (Rf) model.

mainly shows the identified instances for each feature and the impact of each instance to make predictions. To provide a notion of the Shapley value distribution for each feature, overlapping dots are flailed along the y-axis. The characteristics are arranged in ascending order of the impact of features. The analysis reveals that significant variations in the magnitudes of the input parameters enable the models to discriminate between different failure case situations.

The inspection from Fig 15 has depicted the contributions of features to the model's prediction for a particular scenario. The value of the feature is shown on the left, and its contribution to the prediction is indicated on the arrows.

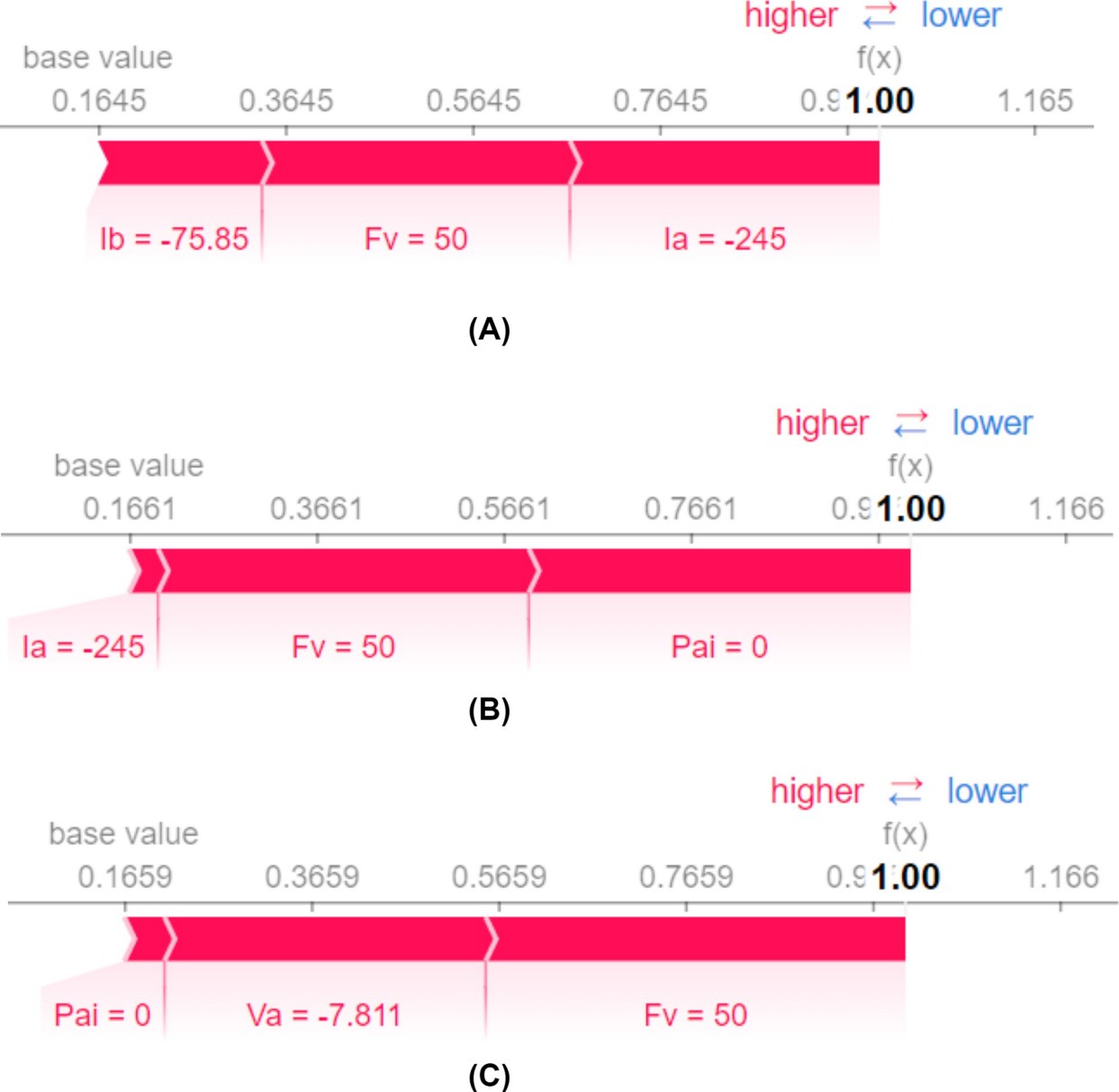

**Fig 15. Explainability of a single observation: K-NN.** (A) Influence of features to predict instance number 8. (B) Influence of features to predict instance number 16. (C) Influence of features to predict instance number 32.

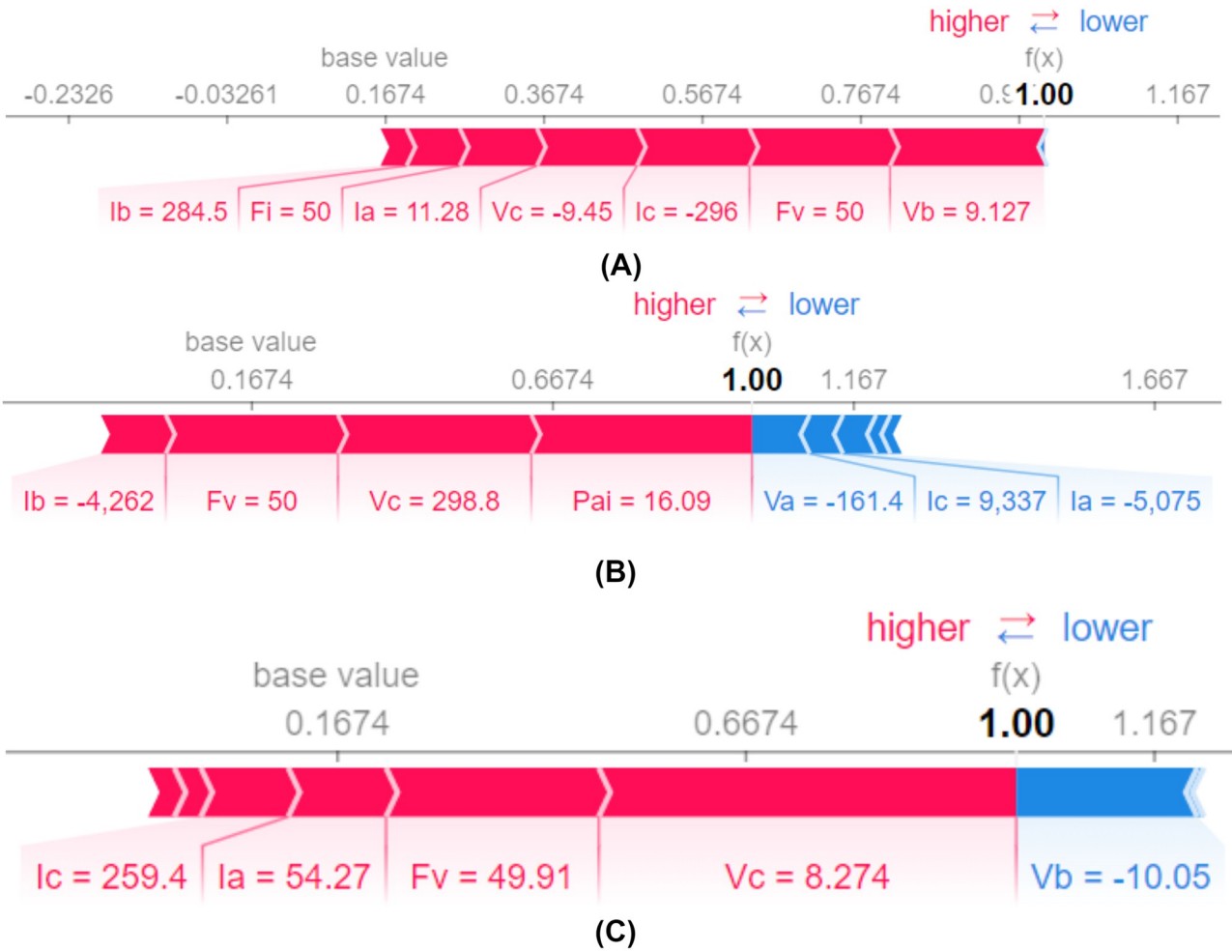

**Fig 16. Explainability of a single observation: DT.** (A) Influence of features to predict instance number 15. (B) Influence of features to predict instance number 30. (C) Influence of features to predict instance number 60.

This representation from Fig 16 has represented the impact of the features in terms of a particular case. To predict a particular circumstance, which features have been involved and how it has contributed to the prediction is visualized by this approach.

Explainable AI has been used to clarify how machine learning models determine the pattern to classify faults from the provided dataset. This approach demonstrates the fact that based on which criteria, each model has been producing outputs. This analysis improves the dependability of these artificial intelligence models. Accordingly, it provides an in depth illustration of background analysis for classifying fault case scenarios.

## Sensitivity analysis

The sensitivity of the developed model to different parameters has been thoroughly analyzed in this section. Different parameters have been retrieved from the Phasor Measurement Unit (PMU) and selected as training inputs to feed into the machine learning models. The effectiveness of the features against each class has been analyzed from top to bottom in this study. Specifically, the identification of actual outcomes from a specific group of features has been shown

**Table 5. Representation of the input groups with specified parameters for sensitivity analysis.**

| Parameter | Group Number | Status |
|---|---|---|
| Ia, Ib, Ic | G1 | Combination of current phases. |
| Fi, Fv, Pai | G2 | Combination of frequencies across current and voltage phases as well as phase angle across current phases. |
| Va, Vb, Vc | G3 | Combination of voltage phases. |
| Ia, Ib, Ic, Fi, Fv, Pai | G4 | Combination of group G1 and G2. |
| Ia, Ib, Ic, Va, Vb, Vc, Fi, Fv, Pai | G5 | Combination of group G1, G2 and G3 which has been selected as the final parameters for training the model. |
| Ia, Ib, Ic, Va, Vb, Vc, Fi, Fv, Pai, Pav, Im, Vm | G6 | Entire set of parameters that has been retrieved from the Phasor Measurement Unit (PMU) including the coo-related parameters. |

in this section. Features have been divided into subgroups to use separately as input to the developed ensemble model.

Initially, the individual group has formed considering similar parameters, which has been presented in Table 5. All the separated groups have been fed into the developed ensemble learning based model sequentially and the outcomes have been noted accordingly.

Initially, three separated groups have been formed considering all the similar parameters then each separated group has been fed to the ensemble model, and the performances have been recorded accordingly. Similarly, combining individual groups one by one and feeding them into the developed ensemble model has ensured the best combination of parameters to obtain the highest performance. Outcomes from the separated groups are presented in Fig 17. It is notable that selected features that have been fixed as the input to train the model have achieved the highest validation accuracy across 10 folds. Similarly, the outcomes from each subgroup and also from combining the subgroups one by one have been presented in this section to analyze the sensitivity of the developed model across different parameters. The performance has increased progressively when a new group of features has been added to the model training. Accordingly, the correlated features that have been removed from the training set have decreased the model validation score. Hence, it is notable that these correlated parameters have affected the predicted performance of the developed model.

## Experiment and result analysis

In this study, short circuit faults have been classified employing supervised techniques using Phasor Measurement Unit (PMU) data. When a short circuit occurs, six labels are retrieved, including no fault, single line to ground fault, double line fault, double line to ground fault, three phase fault, and three phase to ground fault. Multiclass classifiers have been used to categorize the six defined labels.

The classification of transmission line disturbances [22, 53] has been performed using nine distinct supervised multiclass techniques, which have been enlisted in Table 6. Decision Tree (DT) [54], Random Forest (RF) [55], and K-Nearest Neighbor (K-NN) [56, 57] have achieved more than 99% train and test accuracy. Decision Tree (DT), Random Forest (RF), and K-Nearest Neighbor (K-NN) have outperformed the other classifiers in the classification of six labels of short circuit fault scenarios. Hence, an ensemble model [58] has been constructed, which combines the prediction performance of the three individual models including Decision Tree (DT), Random Forest (RF) and K-Nearest Neighbor (K-NN) has achieved 99.99% train accuracy and 99.92% test accuracy. In a similar vein, Gradient Boosting (GB), a well-known classifier for multiclass classification, has attained more than 97% train and test accuracy. The

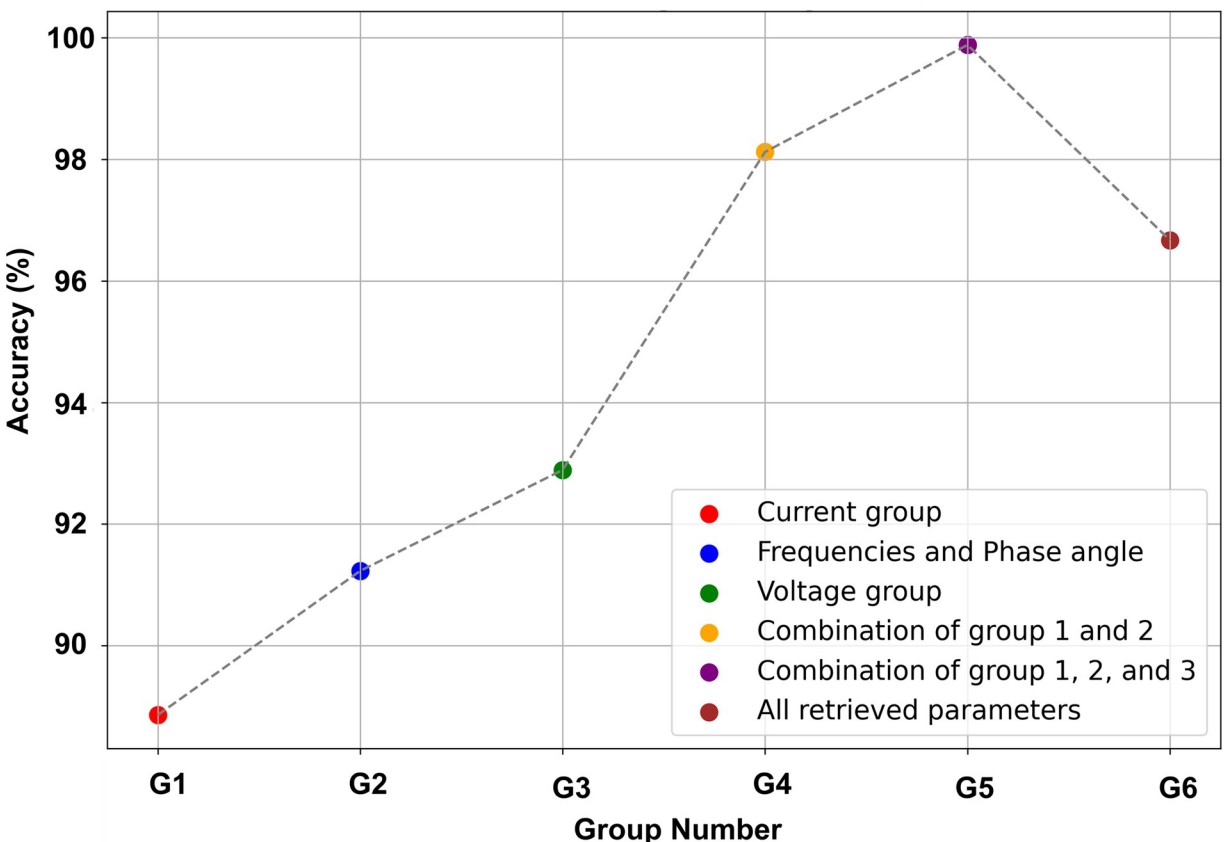

**Fig 17. Illustration of the sensitivity analysis of the proposed ensemble learning based approach across different input parameters.**

Gradient Boosting Classifier has also performed quite effectively. An artificial neural network Multilayer Perceptron (MLP) that employs backpropagation for network training has achieved more than 95% test and train accuracy. Accordingly, Gaussian Naive Bayes (GNB) has also achieved more than 95% of train and test accuracy. The Support Vector Classifier (SVC) method followed by one vs. one approach, which divides the dataset into separate datasets for each label vs all other labels, achieved more than 92% train and test accuracy. Accordingly, the

**Table 6. Result summary of all the applied models.**

| Classifiers | Train Accuracy | Test Accuracy |
| --- | --- | --- |
| EM (DT, RF, K-NN) | 99.99% | 99.92% |
| DT | 99.99% | 99.87% |
| RF | 99.99% | 99.86% |
| K-NN | 99.83% | 99.78% |
| GB | 97.92% | 97.88% |
| MLP | 95.67% | 95.64% |
| GNB | 95.27% | 95.21% |
| SVC One VS One | 92.21% | 92.19% |
| SVC One VS Rest | 86.85% | 86.83% |

One vs. Rest Support Vector Classifier (SVC), which splits the dataset into individual binary datasets for each label, obtained more than 86% train and test accuracy. In comparison to other classifiers, Multilayer Perceptron (MLP), Gaussian Naive Bayes (GNB), and Support Vector Classifier (SVC) in both techniques, One VS One and One Vs Rest have been performed significantly low.

To ascertain if the model leads to overfitting, cross-validation has been performed. Three supervised classification models outscored other models in terms of performance. Cross-validation has been done on these specific well-performing models to identify overfitting issues. Data has been divided into training and testing sets repeatedly, and accuracies are computed using the trained estimator from the training set. The average accuracy score for each of the selected classifiers was determined using cross-validation across 10 folds.

Decision Tree (DT), Random Forest (RF), and K-Nearest Neighbor (K-NN) have achieved more than 99% cross-validation accuracy, which is similar to the test accuracy. Accordingly, the cross-validation accuracy for the Ensemble Model (DT, RF, K-NN) is also more than 99%, which matched the level of its test accuracy. Notably, the cross-validation scores of the selected classifiers have achieved the benchmark of the computed test accuracy. In light of the discussion from Table 7, it can be asserted that constructed classification models do not lead to overfitting.

In terms of classifying different faults, our developed model performed better than the outcomes accumulated by the recent studies in similar fields that have been presented in Table 8. In light of this, it is notable that our model has outperformed by being trained with the parameters that have been retrieved from the Phasor Measurement Unit (PMU).

To gauge the effectiveness of used supervised multiclass classifiers, confusion matrices are implemented, which has been depicted in Fig 18. The confusion matrix compares actual and predicted values. Here, a 6 x 6 matrix is represented for each classifier, where six is the number of classes. In the confusion matrices, six distinct classes have been depicted, including no fault, single line to ground fault, double line fault, double line to ground fault, three phase fault, and three phase to ground fault.

Notably, Decision Tree (DT), Random Forest (RF), and K-Nearest Neighbor (K-NN) have been successful in predicting accurate outputs. In terms of correctly anticipating outputs, the

**Table 7. Cross validation score of the best performing models across 10 folds.**

| Classifiers | Cross Validation Score |
|---|---|
| EM (DT, RF, K-NN) | 99.88% |
| DT | 99.84% |
| RF | 99.83% |
| K-NN | 99.76% |

**Table 8. Performance comparison of the proposed approach to classify faults with existing approaches.**

| Classification Approach | Result |
|---|---|
| **Our model: EM (DT, RF, K-NN)** | **99.88%** |
| CNN [27, 28] | 99.52% |
| LSTM [35, 36] | 99.00% |
| CNSF [34] | 99.00% |
| BiLSTM [37] | 98.13% |
| SVM & PCA [31] | 80% |

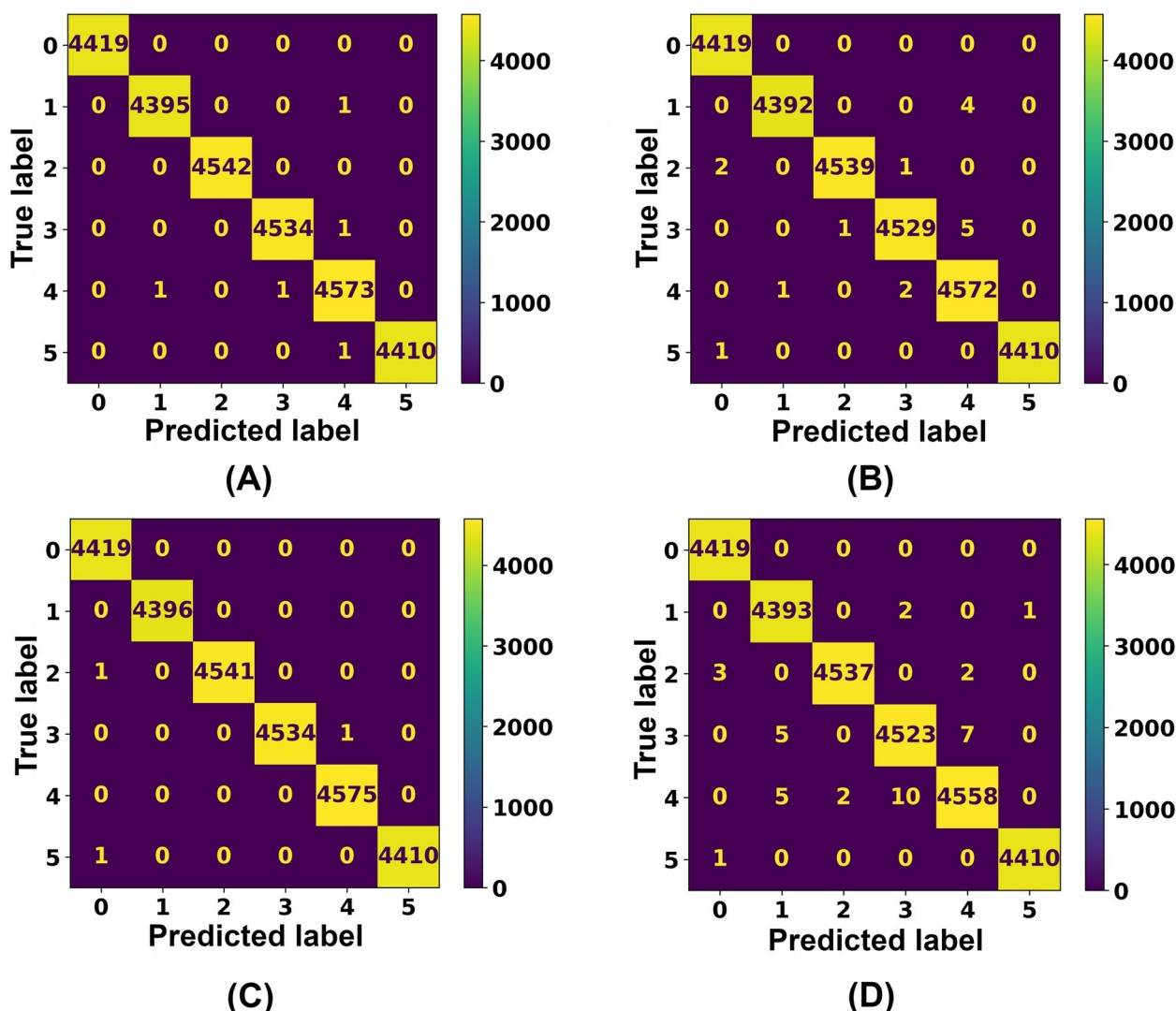

**Fig 18. Representation of confusion matrix for the best performing models.** (A) Confusion matrix for Ensemble (DT, RF, K-NN) model. (B) Confusion matrix for Decision Tree (DT) model. (C) Confusion matrix for Random Forest (RF) model. (D) Confusion matrix for K-Nearest Neighbour (K-NN) model.

constructed Ensemble Model (DT, RF, K-NN) has also performed notably. Most of these classifiers make a small number of wrong predictions in class three and class four, which correspond to the double line to ground (LL-G) fault and three phase (LLL) fault respectively.

The micro average and weighted average values of precision, recall, and f1-score of the applied classifiers have been shown in the classification report. For the classes including "No Fault (NF)", "line to ground fault (L-G)", "line to line fault (LL)", "line to line to ground fault (LL-G)", "line to line to line fault (LLL)" and "line to line to line to ground fault (LLL-G)" respectively 4594, 4452, 4490, 4448, 4445 and 4452 samples are tested. In total 26881 numbers of samples have been used to test the models which is the value of support.

Classification reports for all the applied modeling approaches have been presented in Table 9. Precision, recall and f1-score values of the Decision Tree (DT), Random Forest (RF), and K-Nearest Neighbor (K-NN) represent these classifiers that have outperformed in

**Table 9. Classification report for all applied models.**

| Classifiers | Precision | Recall | F1-score |
|---|---|---|---|
| EM (DT, RF, K-NN) | 1 | 1 | 1 |
| DT | 1 | 1 | 1 |
| RF | 1 | 1 | 1 |
| K-NN | 1 | 1 | 1 |
| GB | 0.98 | 0.98 | 0.98 |
| MLP | 0.96 | 0.96 | 0.96 |
| GNB | 0.96 | 0.95 | 0.95 |
| SVC One VS One | 0.93 | 0.92 | 0.92 |
| SVC One VS Rest | 0.89 | 0.87 | 0.87 |

classifying faults from the given dataset. As a consequence, the constructed Ensemble Model (DT, RF, K-NN) has an outstanding micro and weighted average score for precision, recall, and f1-score. Accordingly, Gradient Boosting (GB) has few wrong predicted values so it has a lower average score for precision, recall, and f1-score. Similarly, Multilayer Perceptron (MLP) and Gaussian Naive Bayes (GNB) have also a lower average score for precision, recall, and f1-score as a result of having a few incorrectly predicted outputs. Finally, the Support Vector Classifier (SVC) in both approaches one vs. one and one vs. rest, have notably lower micro and weighted average scores compared to the other applied classifiers due to having a significant number of incorrect predicted values.

As Decision Tree (DT), Random Forest (RF), and K-Nearest Neighbor (K-NN) classifiers have performed consistently well on this dataset, hyperparameter tuning has been performed on these classifiers using random and grid search approaches to identify for which parameters of these classifiers give the best result. The set of hyperparameters has randomly passed through RandomizedSearchCV for each specified model, which calculates the score and returns the optimal set of hyperparameters with the best result. Similarly, GridSearchCV uses the Cross-Validation approach to analyze the given model for each combination of the parameters provided in the dictionary in order to get the optimal combination of parameters with the highest score. Finally, the performance of these three classifiers with the optimal hyperparameters has been combined in the ensemble model, which has been shown in Fig 19. Notably, every positive and negative class point can be distinguished correctly by the ensemble model.

The receiver operating characteristic (ROC) curve visually depicts the performance of the ensemble model for each classification level. This statistic takes the trade-offs between recall and accuracy into consideration. From Fig 20, it is notable that the ensemble model has performed marvelously in classifying each class. The proposed hybrid model has successfully classified 26878 test samples from six distinct classes as clean and faulty instances in 0.4977 seconds.

To ensure the model's effectiveness the ensemble model has been tested on the IEEE 14 bus system [59]. The developed model has been exported in Matlab Simulink as.mat format and connected the model to the appropriate components of the 14-bus system to receive data as input. In the IEEE 14 bus environment, different fault cases have been simulated using fault circuits on the Matlab Simulink to check if the developed model can classify short circuit fault instances. Additionally, best-performing models have been tested separately on the 14-bus system to determine individual effectiveness. The developed ensemble model has been successful in classifying short circuit instances of the transmission line from the IEEE 14 bus system.

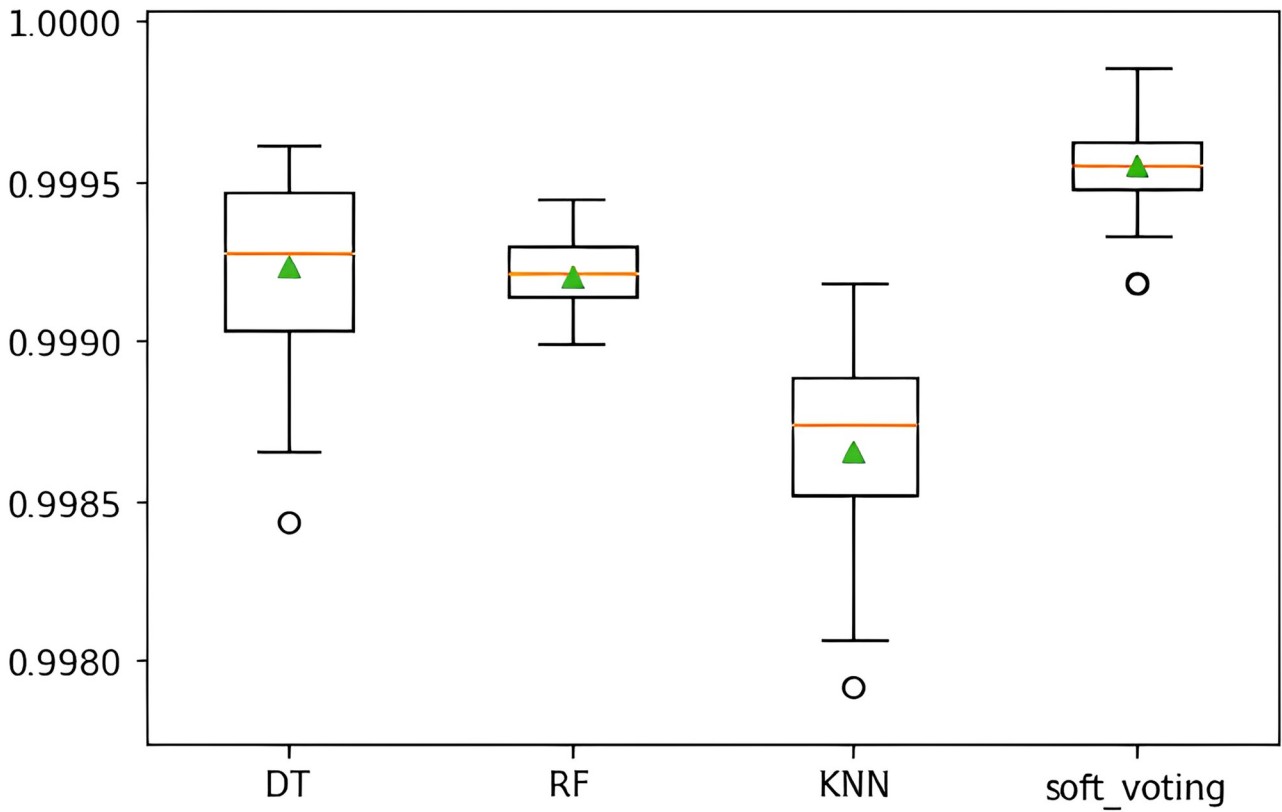

**Fig 19. Visualization of the contribution of each combined model in ensemble learning through the voting report.**

## Conclusion

Retrieving real-time data could be tough for power system-supervised research as it requires labeled datasets to train models. Accordingly, it is not technically easy to introduce faults in any existing power grid to retrieve labeled datasets or it could be really expensive to set a physical environment to test different fault scenarios and collect data with proper labels. Hence, simulating fault case scenarios could be an impactful solution to collect the labeled faulty data. Rui et al. [10] presented a PMU based approach for classification problems. This paper presented the significance of PMU data in classifying power system cyber attack events. This research to classify transmission line short circuit faults has been done using the Phasor Measurement Unit (PMU) data that ensures more features to train the prediction models. The significant impacts of the additional features that could have been possible to retrieve due to using a Phasor Measurement Unit (PMU) have been represented in this research.

It has been noted that Decision Tree (DT), Random Forest (RF), and K-Nearest Neighbor (K-NN) classifiers have outperformed other multiclass classifiers in categorizing different types of faults from the dataset. Considering the output from performance metrics, it has been decided to combine the performance of these classifiers through ensemble learning [58]. Hence, optimal parameters of these classifiers that ensure consistent performance have been used to construct the final classification model. The performance of these consistently effectuating models has been cross-verified through explainable AI [50, 52, 60], which increased the dependability of the classification model.

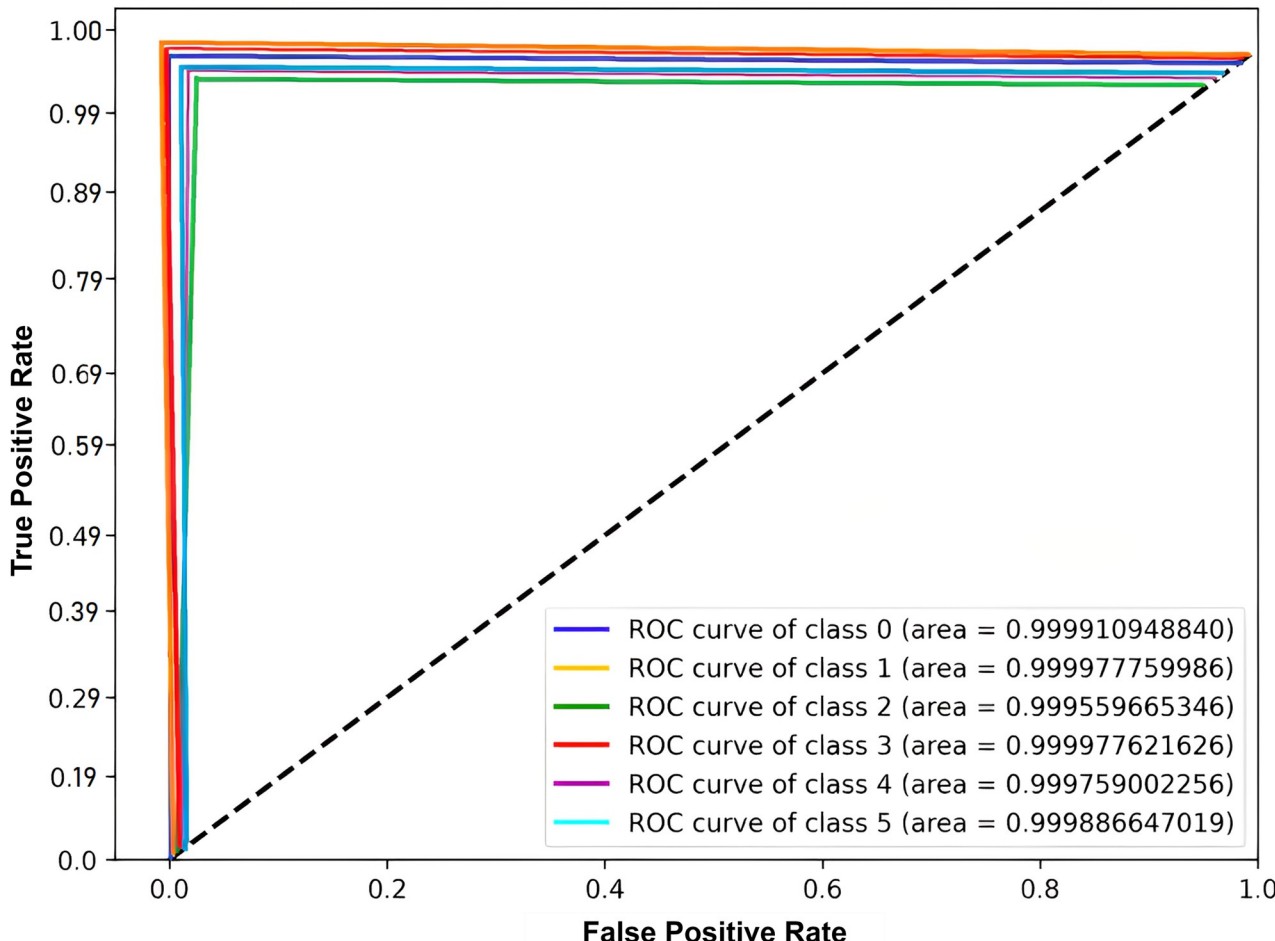

**Fig 20. Representation of the Receiver operating characteristic (ROC) curve for the proposed ensemble approach.**

For now, testing the developed model in terms of a physical setup or real environment hasn't been possible due to the limitation of real time labeled dataset of transmission lines or a complete physical setup to test this model. This boundary is considered the limitation of this research. The real-time data retrieval hardware setup is now being implemented. To make the model more accurate and realistic, the author wants to work on the real-time dataset in the near future.

## Author Contributions

**Conceptualization:** Simon Bin Akter, Tanmoy Sarkar Pias, Shohana Rahman Deeba, Hafiz Abdur Rahman.

**Formal analysis:** Simon Bin Akter, Tanmoy Sarkar Pias, Hafiz Abdur Rahman.

**Funding acquisition:** Hafiz Abdur Rahman.

**Investigation:** Simon Bin Akter, Tanmoy Sarkar Pias.

**Methodology:** Simon Bin Akter, Shohana Rahman Deeba, Hafiz Abdur Rahman.

**Project administration:** Hafiz Abdur Rahman.

**Software:** Simon Bin Akter.

**Supervision:** Shohana Rahman Deeba, Jahangir Hossain, Hafiz Abdur Rahman.

**Validation:** Simon Bin Akter.

**Visualization:** Simon Bin Akter.

**Writing – original draft:** Simon Bin Akter.

**Writing – review & editing:** Tanmoy Sarkar Pias, Shohana Rahman Deeba, Jahangir Hossain, Hafiz Abdur Rahman.

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
