## [Decision Letter · Decision Letter 0]

6 Jun 2023

PONE-D-23-11739Ensemble learning based transmission line fault classification using phasor measurement unit (PMU) data with explainable AI (XAI)PLOS ONE

Dear Dr. Abdur Rahman,

Thank you for submitting your manuscript to PLOS ONE. After careful consideration, we feel that it has merit but does not fully meet PLOS ONE’s publication criteria as it currently stands. Therefore, we invite you to submit a revised version of the manuscript that addresses the points raised during the review process.

We look forward to receiving your revised manuscript.

Kind regards,

Praveen Kumar Donta, Ph.D.

Academic Editor

PLOS ONE

Journal Requirements:

4. Please ensure that you refer to Figures 1-19 in your text as, if accepted, production will need this reference to link the reader to the figure.

5. We note you have included a table to which you do not refer in the text of your manuscript. Please ensure that you refer to Table 1-7 in your text; if accepted, production will need this reference to link the reader to the Table.

Reviewers' comments:

Reviewer's Responses to Questions

**Comments to the Author**

1. Is the manuscript technically sound, and do the data support the conclusions?

Reviewer #1: Yes

Reviewer #2: Partly

Reviewer #3: Yes

2. Has the statistical analysis been performed appropriately and rigorously? 

Reviewer #1: Yes

Reviewer #2: No

Reviewer #3: I Don't Know

3. Have the authors made all data underlying the findings in their manuscript fully available?

Reviewer #1: Yes

Reviewer #2: Yes

Reviewer #3: Yes

4. Is the manuscript presented in an intelligible fashion and written in standard English?

Reviewer #1: Yes

Reviewer #2: Yes

Reviewer #3: Yes

5. Review Comments to the Author

Reviewer #1: Dear authors,

I have read your paper, titled " Ensemble learning based transmission line fault classification using phasor measurement unit (PMU) data with explainable AI (XAI) ", carefully.

I am of the convection that the paper needs some serious revisions before I recommend it for publication in this journal. Please necessarily apply the comments and highlight them in the paper.

The following presents my comments.

1.The Abstract section of the paper has been written very poorly. You are expected to state the necessity of your research and its novelty correctly. Please describe the software implementation and algorithm results briefly. I think the Abstract section needs to be 200-300 words and formatted in a standard way. Please revise the Abstract.

2.The Introduction section has been written acceptably but it is recommended to divide it into five parts as follows:

(a)Research motivation, (b) literature review, (c) the necessity of the research based on challenges of the literature, (d) novelty and main contributions of the paper, and (e) organization and structure of the paper.

3.The output results of the simulation are very brief. In this section, it is better to present more graphical and numerical results in more scenarios that can show the accuracy of the proposed algorithm well. In this section, you should be able to defend your algorithm by providing appropriate simulation results.

4.Sensitivity of the proposed algorithm should be analyzed with respect to fundamental parameters. Please add a new item title named "Sensitivity Analysis" and test the sensitivity of the algorithm to different parameters and conditions and provide the results in numerical and graphical forms.

Reviewer #2: Following are the comments for improvement:

1. Why are the researchers considering values of reactive power in milli i.e., mVA, power in mW?

2. The results are very blurred. It must be improved before taking into consideration for further review.

3. Mathematical model must be added for transmission line under different fault conditions.

4. Ensemble learning is used by authors? Any benefits over advanced learning methods already available in literature.

5. Testing, training and validation is not explained.

6. 10 fold validations? Why any specific reason?

7. Data preprocessing " It is noticeable that in terms of no fault cases value of current and voltage phases are normally distributed." How? and Why? It can be some other distribution as well.

8. The information related to outliers are not explained.

9. The limitations of the research work must be added.

10. RF, KNN, Decision Tree, Ensemble results are not fully added. Some of the results are missing for the mentioned techniques.

11. Correlation matrix ? Whta is the significance of it?

12. Region of Occurrence ? Why? Analysis must be done with supported technical arguments.

13. Comparative table must be added in order to support the proposed method as compared to existing methods.

14. IEEE 14 bus system is mentioned in abstract. No details are added in the research about it.

Reviewer #3: The paper presents an ensemble learning based Transmission line fault detection and classification. Following are my comments.

1. The citation of the references order in the manuscript is random i.e. The first citation begins with a reference[19]. It should be serially in incremental order.

2. Authors are claiming the cutting-edge accuracy at 99.88% on Synthesized data, what is the guarantee that the sysnthesized result will reflect in real-time in physical transmission line.

3. It is suggested to the authors to perform the same analysis based on the data from the physical transmission line or practical laboratory setup, and compare the accuracy with simulated results.

4. In the real-time transmission line the voltage, current and the power in higher scale. The simulated results are performed in Milli volt and Milli watt level, What is the practicality of applying the proposed methodology in physical real-time transmission line.

5. The resolution of some figures is very poor (such as Fig.6, Fig.7, Fig.11, Fig.14) it need to be improved.

6. PLOS authors have the option to publish the peer review history of their article (what does this mean?). If published, this will include your full peer review and any attached files.

Reviewer #1: No

Reviewer #2: No

Reviewer #3: No

---

## [Author Response · Author response to Decision Letter 0]

21 Aug 2023

A separate document is added for “Response to Reviewer” with proper formatting. Following is the unformatted text from that document.

Reviewer #1: Dear authors, I have read your paper, titled " Ensemble learning based transmission line fault classification using phasor measurement unit (PMU) data with explainable AI (XAI) ", carefully. I am of the convection that the paper needs some serious revisions before I recommend it for publication in this journal. Please necessarily apply the comments and highlight them in the paper.

The following presents my comments.

1. The Abstract section of the paper has been written very poorly. You are expected to state the necessity of your research and its novelty correctly. Please describe the software implementation and algorithm results briefly. I think the Abstract section needs to be 200-300 words and formatted in a standard way. Please revise the Abstract.

Thank you for bringing up this fact. We appreciate your acknowledgment. We have revised our Abstract section and this time we have illustrated the necessity and novelty of our research more specifically. Additionally, the data generation process, software implementation, and the outcomes from our approach have been discussed briefly in the Abstract section. 

2. The Introduction section has been written acceptably but it is recommended to divide it into five parts as follows: (a)Research motivation, (b) literature review, (c) the necessity of the research based on challenges of the literature, (d) novelty and main contributions of the paper, and (e) organization and structure of the paper.

We appreciate your acknowledgment. We have divided the Introduction section into five parts sequentially including Research motivation, Literature review, Necessity of the research, Novelty and major contributions, and Organization and structure according to the given recommendation and described each subsection with appropriate arguments. 

3. The output results of the simulation are very brief. In this section, it is better to present more graphical and numerical results in more scenarios that can show the accuracy of the proposed algorithm well. In this section, you should be able to defend your algorithm by providing appropriate simulation results.

We appreciate your acknowledgment. We have added a new table, Table 8 in Experiment and Result Analysis section to compare our fault classification technique with the existing approaches. This comparison supports the effectiveness of our model to classify faults as our model have been performed better than the outcomes achieved from recent studies in similar fields. However, in the Experiment and Result Analysis section, we have represented the analysis of the results thoroughly: 

i. In Table 5, train and test accuracy for all the tested models have been represented. We have conducted a comparison of all the tested models based on their achieved accuracy scores.

ii. To check if the model is overfitted or not, we have presented cross-validation scores for best-performing models in Table 6. 

iii. In Fig 17, To illustrate the performance in terms of each class, confusion matrices for the best-performing models have been presented that gives a comparison between actual and predicted values. 

iv. Classification reports have been presented in Table 6 which provides precision, recall, and f1-score for all the tested models. It ensures the evaluation of the models more preciously. 

v. As we have performed ensemble learning and combined the predicted performances of best-performing models through soft voting, the voting report to show the contribution of each model to increase the predicted performance of the final model has been visualized in Fig 18.

vi. Receiver operating characteristic (ROC) curve for multiclass has been presented in Fig 19 for the ensemble model to evaluate the performance of the ensemble model against each class. 

We have evaluated the developed model with different performance matrices to support the effectiveness of our developed model in Experiment and Result Analysis section. We have compared different classifier outcomes to identify the best-performing models. Performance against each class has been shown for the individual best-performing models and also for the combined model accordingly. 

4. Sensitivity of the proposed algorithm should be analyzed with respect to fundamental parameters. Please add a new item title named "Sensitivity Analysis" and test the sensitivity of the algorithm to different parameters and conditions and provide the results in numerical and graphical forms.

Thank you for addressing this question. With the aim of enhancing comprehensibility for readers, we have added a new section Sensitivity Analysis with Table 5 and Fig 17 to our research. We divided the dataset into multiple subgroups and then used these subgroups both individually and combined with other subgroups as the input to the train ensemble model. Finally, we have compared the obtained outcomes to check the sensitivity of the developed model in terms of different parameters. 

Reviewer #2: Following are the comments for improvement:

1. Why are the researchers considering values of reactive power in milli i.e., mVA, power in mW?

Thank you for bringing up this particular matter. We deeply regret to inform you that an error occurred during the writing process, resulting in a misinterpretation of the labeling format for reactive power. We sincerely apologize for any confusion or inconvenience this may have caused. We have rectified this error and revised the labeling in the Data Generation section. Specifically, we have replaced the previous labeling with the appropriate units of MVA (mega volt-ampere) and MW (megawatt) in order to accurately represent the reactive power measurements.

2. The results are very blurred. It must be improved before taking into consideration for further review.

Thank you for your effective identification. All the figures have regenerated and this time figures' dots per inch (DPI) have been set to 300 also quality has increased through AI tools for specific pictures including Figure 11, Figure 13, Figure 14, Figure 15, Figure 16, Figure 20, and Figure 20. The instructions specifying the areas that require improvement, will be more effective for us to make changes.

3. Mathematical model must be added for transmission line under different fault conditions.

We appreciate your acknowledgment. However, the mathematical model will be inappropriate for our research as we proposed a completely different approach to classify transmission line faults. The mathematical model is an equivalent circuit representation of a transmission line, consisting of series impedance (Z), shunt admittance (Y), and input/output nodes. It helps analyze transmission line behavior under different conditions by manipulating the circuit. On the other hand, our approach is an artificial intelligence (AI) based model that can classify different faults by identifying a pattern through analyzing the historical data of different fault conditions. We have trained different machine learning multiclass classifiers using the historical faulty labeled data of transmission lines and identified different transmission line faults based on the given parameters. To sum up, it can be said these two approaches can be considered as substitutes for one another. Hence, the implementation of the mathematical model for each fault type won’t be appropriate for our research. 

4. Ensemble learning is used by authors? Any benefits over advanced learning methods already available in literature.

We appreciate your acknowledgment. However, we have represented how our fault classification approach has performed better than the existing approaches in similar fields in the Experiment and Result Analysis (Table 8) section. We have retrieved a multivariate dataset to classify faults and identified nine features as the ideal input after pre-processing the dataset. Our intention is to extract features from phasor measurement (PMU) data through exploratory data analysis (EDA) as we can show the effectiveness of the phasor measurement unit (PMU) data to classify transmission line faults, which is a vital factor of our research. Hence, we used machine learning techniques to address this issue, which has enabled us to extract features through data analysis not with the help of convolution layer piles and sets of pooling layers. Our approach ensures more interpretability and explainability compared to other approaches by providing more transparent and interpretable models, enabling a better understanding of how the input features influence the output. In light of this, we can say that our approach requires explicit feature engineering, where domain knowledge is utilized to extract relevant features from the data. This can be advantageous when there is prior knowledge about the problem domain and specific features that are meaningful for the task at hand. Additionally, the dataset that we have generated performed reasonably well in the machine learning algorithms rather than other approaches. The three machine learning models outperformed to predict transmission line faults, as well as these models, performed considerably well in IEEE 14 bus systems since we have made the decision to combine the performance of these models through soft voting based on knowledge gathered from the literature review, which represented how the effect of simulation tests in this field reflect in real life. Literature review related to this field has shown evaluation through simulation is considerably reliable so we can rely that the combined performance of the three models can handle real-life scenarios as well. 

5. Testing, training and validation is not explained.

We appreciate your acknowledgment. However, in the Data Preparation section, we have explained the details of the train and test set which includes the train-test splitting, count of data in the train-test set, and label encoding of the target feature. In short, the workflow of data preparation from separating the inputs and target to feeding it to the machine learning models has been visualized in the Data Preparation section. In Table 5 of the Experiment and Result Analysis section, both train and test accuracy for all the tested models has been interpreted. Additionally, in Table 6 of the Experiment and Result Analysis section, the cross-validation score has been interpreted for best-performing models. 

6. 10 fold validations? Why any specific reason?

Thank you for addressing this question. With the aim of enhancing comprehensibility for users, we have added the reason for using 10-fold in the Methodology section. The 10-fold cross-validation technique has been employed in this study to ensure an ample amount of data is available for training and validation purposes. This method divides the dataset into ten equal-sized subsets or "folds." During each fold, 90% of the data is utilized for training the model, while the remaining 10% is allocated for validation. This division allows for a substantial training dataset, which enhances the model's ability to learn effectively. Moreover, the validation set in each fold is sufficiently large, enabling a reliable estimation of the model's performance. By utilizing 10-fold cross-validation, we have achieved a more stable assessment of the model's performance compared to using a smaller number of folds. Similarly, the utilization of an increased number of folds in the cross-validation process has resulted in a notable reduction in the sensitivity of the estimated performance to the particular partitioning of the data. 

7. Data preprocessing " It is noticeable that in terms of no fault cases value of current and voltage phases are normally distributed." How? and Why? It can be some other distribution as well.

We appreciate your acknowledgment regarding the reference made to this matter. We have conducted the distribution check to identify outliers in the data. As in no-fault conditions, data (current & voltage) are symmetrically distributed with no unexpected skew. Additionally, we have calculated skew and kurtosis values for the no-fault condition, which is in the -2 to 2 and -7 to 7 range respectively. These findings indicate that the data are normally distributed during the no-fault condition. We have discussed these facts in the Preprocessing section. 

8. The information related to outliers are not explained.

Thank you for addressing this question. We appreciate your acknowledgment regarding the reference made to this matter. We have added a discussion of the technique that is used to handle the outliers in the Preprocessing section. We have used the interquartile range (IQR) technique that removes the data points outside the range of 1.5 times. Any observations that have been found more than 1.5 IQR below Q1 or more than 1.5 IQR above Q3 have been considered outliers and removed from the dataset. 

9. The limitations of the research work must be added.

Thank you for addressing this question. We have mentioned the limitation of this research in the Conclusion section. As for now, the physical setup to test this model hasn't been ready as well as real-time labeled data of the transmission lines is also limited, so our developed model has only been tested on simulation-based platforms. This boundary has been considered the limitation of this research. 

10. RF, KNN, Decision Tree, Ensemble results are not fully added. Some of the results are missing for the mentioned techniques.

We appreciate your acknowledgment. We have added a new section Sensitivity Analysis with Table 5 and Fig 17 to demonstrate the obtained outcomes from similar groups of features. In this section, we have used separate groups of parameters individually and also combined them one by one with other groups as the input of the developed model to analyze the sensitivity of the ensemble model in terms of different parameters. This section represents the model outcomes obtained by the developed ensemble model more enormously. However, in the Experiment and Result Analysis section, we have thoroughly represented the analysis of the results obtained from Decision Tree, RF, KNN, and Ensemble learning: 

i. In Table 5, train and test accuracy for all the tested models have been represented. We have conducted a comparison of all the tested models based on their achieved accuracy scores and identified DT, RF, KNN, and the Ensemble model as the best-performing model.

ii. To check if these selective models (DT, RF, KNN, and Ensemble) are overfitted or not, we have presented cross-validation scores across 10 fold in Table 6. 

iii. In Fig 17, To illustrate the performance in terms of each class, confusion matrices for the best-performing models (DT, RF, KNN, and Ensemble) have been presented that give a comparison between actual and predicted values. 

iv. Classification reports have been presented in Table 6 which provides precision, recall, and f1-score for all the tested models including DT, RF, KNN, and Ensemble. It ensures the evaluation of the models more preciously. 

v. As we have performed ensemble learning and combined the predicted performances of best-performing models through soft voting, the voting report to show the contribution of each model to increase the predicted performance of the final model has been visualized in Fig 18.

vi. Receiver operating characteristic (ROC) curve for multiclass has been presented in Fig 19 for the ensemble model to evaluate the performance of the ensemble model against each class. 

We have evaluated these selective (DT, RF, KNN, and Ensemble) models with different performance matrices to support the effectiveness of our developed models in the Experiment and Result Analysis section. We have compared different classifier outcomes to identify the best-performing models. Performance against each class has been shown for the individual best-performing models (DT, RF, and KNN) and also for the combined model (Ensemble Model) accordingly.

11. Correlation matrix ? Whta is the significance of it?

We appreciate your acknowledgment regarding the reference made to this matter. We have discussed the significance of the correlation matrix in the Feature Extraction section. We have identified 12 features from the phasor measurement unit (PMU). However, we have considered only 9 features as the input since there exist correlations among certain input features, which has a significant possibility to generate biased and wrong predictions for the model. Hence, we checked the impact of each feature and keep the most impactful features from the correlated features. Additionally, we have added a new section Sensitivity Analysis (Fig 17) in our research, which cross-verified the fact that correlated features have affected the predicting performance of the developed model. 

12. Region of Occurrence ? Why? Analysis must be done with supported technical arguments.

Thank you for addressing this question. We appreciate your acknowledgment regarding the reference made to this matter. We have added a subsection 0.5 Organization and structure under the Introduction section that demonstrates where we can implement the proposed model and how it can affect the system. The primary goal of this research is to develop an artificial intelligence platform that utilizes Fourth Industrial Revolution (4IR) technologies for the purpose of monitoring the power grid. The proposed model aims to possess the ability to accurately identify and categorize short circuit events in various real-world scenarios. To achieve this, it is intended to deploy the developed model at different locations along the physical transmission line. This deployment strategy will enable the acquisition of comprehensive information regarding any instances of blackouts or other anomalous events that may transpire within the electrical grid.

13. Comparative table must be added in order to support the proposed method as compared to existing methods.

Thank you for addressing this question. We appreciate your acknowledgment regarding the reference made to this matter. We have added a new table, Table 8 in Experiment and Result Analysis section to compare our fault classification technique with the existing approaches. This comparison supports the effectiveness of our model to classify faults as our model have been performed better than the outcomes achieved from the recent studies in similar fields.

14. IEEE 14 bus system is mentioned in abstract. No details are added in the research about it.

Thank you for addressing this question. We have added a demonstration of the evaluation process through the IEEE 14-bus system at the end of the Experiment and Result Analysis section. As IEEE 14 bus is a popular IEEE benchmark bus system so we didn’t explain the components of the bus system. However, we discussed how we have used the 14-bus to evaluate our model. We have exported our ensemble model within the IEEE 14 bus system in .mat format to assess its capabilities. In order to evaluate its predictive accuracy, we generated various fault scenarios using fault circuits and utilized our developed model to predict these scenarios. Additionally, we conducted individual tests of the best-performing models on the 14-bus system to determine their individual effectiveness.

Reviewer #3: The paper presents an ensemble learning based Transmission line fault detection and classification. Following are my comments.

1. The citation of the references order in the manuscript is random i.e. The first citation begins with a reference[19]. It should be serially in incremental order.

Thank you for addressing this question. We appreciate your acknowledgment regarding the reference made to this matter. We have rearranged the references in incremental order. We sincerely apologize for any confusion or inconvenience this may have caused. 

2. Authors are claiming the cutting-edge accuracy at 99.88% on Synthesized data, what is the guarantee that the sysnthesized result will reflect in real-time in physical transmission line.

We appreciate your acknowledgment regarding the reference made to this matter. To validate the performance of our model, conducting physical experiments in a controlled environment is currently unfeasible due to the extensive equipment setup required. As an alternative, we have deployed our ensemble model within the IEEE 14 bus system to assess its capabilities. In order to evaluate its predictive accuracy, we generated various fault scenarios using fault circuits and utilized our developed model to predict these scenarios. Additionally, we conducted individual tests of the best-performing models on the 14-bus system to determine their individual effectiveness. It is worth noting that within the realm of research in this field, evaluating AI models in real-world settings poses significant challenges. Consequently, many studies outline the evaluation of their developed models using IEEE benchmark bus systems. We have demonstrated the evaluation process through the 14-bus system in the Experiment and Result Analysis section. 

3. It is suggested to the authors to perform the same analysis based on the data from the physical transmission line or practical laboratory setup, and compare the accuracy with simulated results.

We appreciate your acknowledgment. Unfortunately, due to the unavailability of labeled real-time data pertaining to transmission line fault cases, we are unable to conduct testing of our model using data from a physical transmission line or a practical laboratory setup. However, we have rigorously evaluated and validated the performance of our model through simulation scenarios. We have demonstrated the evaluation process through the simulation in the Experiment and Result Analysis section. This approach holds a consistent validation method utilized in numerous studies within this field. Nevertheless, we have added this fact as the limitation of this research in the Conclusion section. 

4. In the real-time transmission line the voltage, current and the power in higher scale. The simulated results are performed in Milli volt and Milli watt level, What is the practicality of applying the proposed methodology in physical real-time transmission line.

We appreciate your acknowledgment regarding the reference made to this matter. During the writing process, there was a misinterpretation of the labeling format for reactive power. We sincerely apologize for any confusion or inconvenience this may have caused. We have rectified this error and revised the labeling in the Data Generation section. Specifically, we have replaced the previous labeling with the appropriate units of MVA (mega volt-ampere) and MW (megawatt) in order to accurately represent the reactive power measurements.

5. The resolution of some figures is very poor (such as Fig.6, Fig.7, Fig.11, Fig.14) it need to be improved.

Thank you for your effective identification. All the figures have regenerated and this time figures' dots per inch (DPI) have been set to 300 also quality has increased through AI tools for specific pictures including Figure 11, Figure 13, Figure 14, Figure 15, Figure 16, Figure 20, and Figure 20. 

6. PLOS authors have the option to publish the peer review history of their article (what does this mean?). If published, this will include your full peer review and any attached files.

No, I want my identity to remain anonymous.

---

## [Decision Letter · Decision Letter 1]

8 Sep 2023

PONE-D-23-11739R1Ensemble learning based transmission line fault classification using phasor measurement unit (PMU) data with explainable AI (XAI)PLOS ONE

Dear Dr. Abdur Rahman,

Thank you for submitting your manuscript to PLOS ONE. After careful consideration, we feel that it has merit but does not fully meet PLOS ONE’s publication criteria as it currently stands. Therefore, we invite you to submit a revised version of the manuscript that addresses the points raised during the review process.

We look forward to receiving your revised manuscript.

Kind regards,

Praveen Kumar Donta, Ph.D.

Academic Editor

PLOS ONE

Reviewers' comments:

Reviewer's Responses to Questions

**Comments to the Author**

1. If the authors have adequately addressed your comments raised in a previous round of review and you feel that this manuscript is now acceptable for publication, you may indicate that here to bypass the “Comments to the Author” section, enter your conflict of interest statement in the “Confidential to Editor” section, and submit your "Accept" recommendation.

Reviewer #1: (No Response)

Reviewer #2: All comments have been addressed

Reviewer #3: All comments have been addressed

2. Is the manuscript technically sound, and do the data support the conclusions?

Reviewer #1: Yes

Reviewer #2: Partly

Reviewer #3: Yes

3. Has the statistical analysis been performed appropriately and rigorously? 

Reviewer #1: Yes

Reviewer #2: N/A

Reviewer #3: Yes

4. Have the authors made all data underlying the findings in their manuscript fully available?

Reviewer #1: Yes

Reviewer #2: Yes

Reviewer #3: Yes

5. Is the manuscript presented in an intelligible fashion and written in standard English?

Reviewer #1: Yes

Reviewer #2: (No Response)

Reviewer #3: Yes

6. Review Comments to the Author

Reviewer #1: Dear authors,

I have carefully read the revised version of your paper titled "X". I believe that most of ambiguities and concerns have successfully been addressed by you and the quality of the paper has been improved. I have recommended the acceptance of the paper to the editor of the paper. I congratulate you on this beforehand.

Good luck,

Reviewer #2: The manuscript is improved. However, following issues must be addressed.

1. However, authors must address the issue of time required for classification of faults.

2. The figures still require improvements.

3. An analysis should also be added to verify that the PMUs with this approach work exactly the same for distribution line and transmission lines so that this could be applicable to any power lines.

4. The effect of synchronous condenser seems neglected on fault classification in IEEE 14 bus system.

Reviewer #3: The authors have addressed all my comments. The paper can be accepted at the present form.

The authors have addressed all my comments. The paper can be accepted at the present form.

The authors have addressed all my comments. The paper can be accepted at the present form.

The authors have addressed all my comments. The paper can be accepted at the present form.

7. PLOS authors have the option to publish the peer review history of their article (what does this mean?). If published, this will include your full peer review and any attached files.

Reviewer #1: No

Reviewer #2: No

Reviewer #3: No

---

## [Author Response · Author response to Decision Letter 1]

22 Sep 2023

The following text is also available in the attached Reviewer_answer_2nd.docx file with proper formatting (answer to the second review phase).

Reviewer #2: The manuscript is improved. However, following issues must be addressed.

1. However, authors must address the time required for classification of faults.

Thank you for addressing this matter promptly. I have added this information in the Experiment and result analysis section under the description of Fig 20. 26878 test samples, belonging to six distinct classes, were classified as clean and faulty instances in 0.4977 seconds by the proposed hybrid model.

2. The figures still require improvements.

Thank you for your kind consideration of this issue. The overall figure quality has seen significant improvement, with particular attention given to enhancing the clarity and resolution of images considering “figure text font size must match the paper body font size considerably”. All figures have been incorporated into the paper and their quality has been appropriately adjusted to meet the standards this time. While certain libraries were employed to gain deeper insights into the study through data analysis and explainable AI techniques, it was noted that the initial image outputs from these libraries did not meet the desired level of quality. However, through the application of advanced AI tools, the image quality has been successfully elevated to its highest standard. It is worth highlighting that meticulous care was taken to ensure that all generated images adhere to the stipulated standard, with a uniform DPI setting of 300 across the board. 

3. An analysis should also be added to verify that the PMUs with this approach work exactly the same for distribution line and transmission lines so that this could be applicable to any power lines.

I appreciate your prompt attention to this matter, as it enhances the comprehensiveness of the research. In distribution systems, PMUs are often referred to as "Micro PMUs" or "µPMUs". These devices are designed for use in low-voltage distribution networks, which are the parts of the grid that deliver electricity directly to end-users. However, in transmission systems, standard PMUs are commonly used, and they are not typically referred to as "Micro PMUs". These devices are designed for use in high-voltage transmission networks, which transport electricity over long distances. Distribution line PMUs are typically deployed at lower voltage levels and are used for monitoring and control of distribution system parameters, such as voltage, current, and power quality. Transmission line PMUs are used for monitoring and controlling parameters in high-voltage transmission lines, such as phase angles, frequency, voltage magnitudes, and line impedance. The PMUs used in distribution networks (Micro PMUs) and transmission networks (standard PMUs) are fundamentally different devices designed for different voltage levels and purposes, and their data parameters are not the same. Therefore, we are not claiming that the approach works exactly the same for distribution lines and transmission lines. However, we think this can be a good future direction based on our current research on transmission line fault classification using ML approaches.

4. The effect of synchronous condenser seems neglected on fault classification in IEEE 14 bus system.

I sincerely appreciate your attention to this matter, as it contributes to a more comprehensive examination. In Transmission Line Short Circuit Fault Analysis using Machine Learning, the synchronous condenser or synchronous motor effect is not a primary focus. Typically, fault analysis in transmission lines focuses on identifying and classifying different types of faults, such as short circuits or line-to-ground faults, based on frequencies, phases, voltage, and current measurements. While the synchronous condenser or motor effect does influence the transient behavior of the power system during faults, it is generally a well-understood and predictable phenomenon. Accordingly, the complexity introduced by modeling the synchronous condenser or motor effect does not provide significant benefits for the specific task of fault analysis using machine learning. Machine learning models are designed to learn patterns and anomalies in the data to identify faults accurately. Hence, adding the complexity of modeling synchronous machines does not substantially improve the model's performance for fault detection and classification. These models primarily rely on provided measures including frequency, phase angle, voltage, and current data without delving deeply into synchronous machine dynamics. It has been observed the synchronous condenser or synchronous motor effect is an important consideration in power system analysis and design, but it is not a primary focus in the specific context of machine learning-based Transmission Line Short Circuit Fault Analysis.

---

## [Decision Letter · Decision Letter 2]

6 Nov 2023

PONE-D-23-11739R2Ensemble learning based transmission line fault classification using phasor measurement unit (PMU) data with explainable AI (XAI)PLOS ONE

Dear Dr. Abdur Rahman,

Thank you for submitting your manuscript to PLOS ONE. After careful consideration, we feel that it has merit but does not fully meet PLOS ONE’s publication criteria as it currently stands. Therefore, we invite you to submit a revised version of the manuscript that addresses the points raised during the review process.

We look forward to receiving your revised manuscript.

Kind regards,

Praveen Kumar Donta, Ph.D.

Academic Editor

PLOS ONE

Journal Requirements:

Reviewers' comments:

Reviewer's Responses to Questions

**Comments to the Author**

1. If the authors have adequately addressed your comments raised in a previous round of review and you feel that this manuscript is now acceptable for publication, you may indicate that here to bypass the “Comments to the Author” section, enter your conflict of interest statement in the “Confidential to Editor” section, and submit your "Accept" recommendation.

Reviewer #2: All comments have been addressed

2. Is the manuscript technically sound, and do the data support the conclusions?

Reviewer #2: Yes

3. Has the statistical analysis been performed appropriately and rigorously? 

Reviewer #2: N/A

4. Have the authors made all data underlying the findings in their manuscript fully available?

Reviewer #2: Yes

5. Is the manuscript presented in an intelligible fashion and written in standard English?

Reviewer #2: Yes

6. Review Comments to the Author

Reviewer #2: The manuscript is much improved. The protection time required for different type of faults must be tabulated for extensive understanding.

7. PLOS authors have the option to publish the peer review history of their article (what does this mean?). If published, this will include your full peer review and any attached files.

Reviewer #2: No

---

## [Author Response · Author response to Decision Letter 2]

12 Nov 2023

(PLEASE NOTE: A properly formatted answer given below is available in the document "Response to Reviewers.docx", which has already been submitted)

Hafiz Abdur Rahman, Ph.D.

Professor, North South University

8 November, 2023

Praveen Kumar Donta, Ph.D.

Academic Editor

PLOS ONE

Dear Dr. Kumar Donta,

Re: PONE-D-23-11739R2

Title: Ensemble learning based transmission line fault classification using phasor measurement unit (PMU) data with explainable AI (XAI)

I would like to express my sincere gratitude for taking the time to review our manuscript and for providing valuable feedback. We appreciate the constructive comments and suggestions made by the editor and reviewers.

Journal Requirements:

The reference list has undergone a thorough review and the DOI (Digital Object Identifier) for references numbered 46, 52, and 53 has been revised and updated. Within the literature review section, there has been an update to reference number 31 in the summary table. Additionally, a new reference has replaced the content previously associated with reference number 59.

Reviewers' comments:

Reviewer #2: The manuscript is much improved. The protection time required for different type of faults must be tabulated for extensive understanding.

I appreciate the reviewer's feedback on incorporating protection time requirements based on the control domain of power systems for different types of transmission line faults. While this is indeed a valuable aspect of power system operation and protection, it is essential to clarify that the primary objective of our research is to develop a fault classification framework within the monitoring domain. As such, my work focuses on the accurate identification and categorization of transmission line faults, which can significantly aid in the early detection and diagnosis of faults, supporting overall system reliability and maintenance. 

Transmission line fault classification using machine learning (ML) and the determination of protection time requirements and relay-based control are distinct components within power system analysis. Fault classification with ML aims to enhance early fault detection and diagnosis through data-driven models, primarily serving the monitoring domain, and facilitating improved situational awareness. In contrast, protection time requirements and relay-based control, situated in the control domain, focus on designing protective relays and schemes to ensure swift and selective fault isolation, reducing system disturbance duration and safeguarding equipment. These two areas address separate aspects of power system engineering, with fault classification primarily enhancing monitoring capabilities, while protection time requirements and relay-based control are pivotal for maintaining grid reliability and minimizing damage.

As refference: https://doi.org/10.1016/j.engappai.2021.104504, https://doi.org/10.1063/5.0108588. These articles illustrate the significant role of machine learning in the identification of faults within power systems. 

The implementation of protection time requirements and relay-based control is undoubtedly an important topic in the power domain, but I intend to consider it as a potential avenue for future research, as it falls outside the scope of the current study.

Sincerely, 

Hafiz Abdur Rahman, Ph.D.

hafiz.rahman@northsouth.edu

---

## [Editor Report · Decision Letter 3]

16 Nov 2023

Ensemble learning based transmission line fault classification using phasor measurement unit (PMU) data with explainable AI (XAI)

PONE-D-23-11739R3

Dear Dr. Abdur Rahman,

We’re pleased to inform you that your manuscript has been judged scientifically suitable for publication and will be formally accepted for publication once it meets all outstanding technical requirements.

Kind regards,

Praveen Kumar Donta, Ph.D.

Academic Editor

PLOS ONE
---

## [Editor Report · Acceptance letter]

21 Nov 2023

PONE-D-23-11739R3 

Ensemble learning based transmission line fault classification using phasor measurement unit (PMU) data with explainable AI (XAI) 

Dear Dr. Abdur Rahman:

I'm pleased to inform you that your manuscript has been deemed suitable for publication in PLOS ONE. Congratulations! Your manuscript is now with our production department. 

Kind regards, 

on behalf of

Dr. Praveen Kumar Donta 

Academic Editor

PLOS ONE